# IllumiCraft: Unified Geometry and Illumination Diffusion for Controllable Video Generation

**Yuanze Lin**♣  **Yi-Wen Chen**♮  **Yi-Hsuan Tsai**◇
**Ronald Clark**♣  **Ming-Hsuan Yang**♠♡

♣University of Oxford  ♠UC Merced  ♮NEC Labs America  ◇Atmanity Inc.  ♡Google DeepMind
Project Page: https://yuanze-lin.me/IllumiCraft_page

Input Video  Background  Relit Video Frames

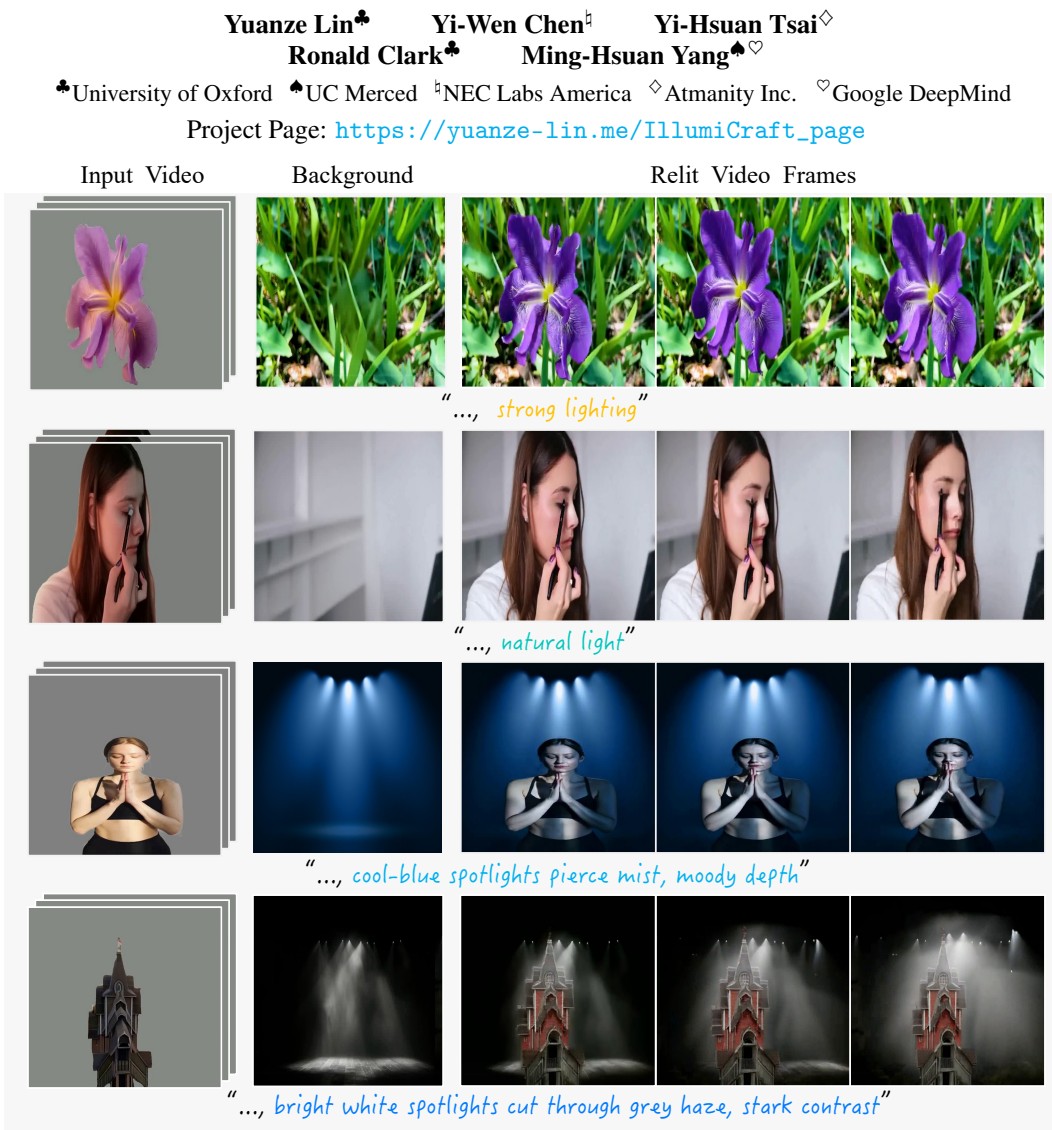

"..., *strong lighting*"

"..., *natural light*"

"..., *cool-blue spotlights pierce mist, moody depth*"

"..., *bright white spotlights cut through grey haze, stark contrast*"

Figure 1: Given a prompt and input video, IllumiCraft edits scene illumination conditioned on the static background image. It handles a variety of illumination scenarios, including spotlight effects.

## Abstract

Although diffusion-based models can generate high-quality and high-resolution video sequences from textual or image inputs, they lack explicit integration of geometric cues when controlling scene lighting and visual appearance across frames. To address this limitation, we propose IllumiCraft, an end-to-end diffusion framework accepting three complementary inputs: (1) high-dynamic-range (HDR) video

39th Conference on Neural Information Processing Systems (NeurIPS 2025).

maps for detailed lighting control; (2) synthetically relit frames with randomized illumination changes (optionally paired with a static background reference image) to provide appearance cues; and (3) 3D point tracks that capture precise 3D geometry information. By integrating the lighting, appearance, and geometry cues within a unified diffusion architecture, IllumiCraft generates temporally coherent videos aligned with user-defined prompts. It supports background-conditioned and text-conditioned video relighting and provides better fidelity than existing controllable video generation methods.

# 1 Introduction

Illumination plays an important role in creating appealing visuals as it helps accentuate the three-dimensional structure of scenes in otherwise flat 2D images. For example, an apple in the sun typically has a bright specular highlight on the side where the surface normal most directly faces the light source, while the far side has a dark soft shadow. This accentuates both the roundness and the glossy texture of the apple. However, despite the importance of this light-geometry interaction, most video generation models [1, 2, 3] treat illumination as an uncontrollable implicit factor.

Achieving coherent relighting in videos involves two main challenges: maintaining consistent illumination over time and ensuring physically plausible light-scene interactions. Light sources should not change abruptly between frames, as this introduces distracting flicker. In addition, shadows, highlights, and reflections need to move consistently with the camera and object motion. Traditional inverse-rendering techniques [4, 5, 6, 7] decompose scenes into albedo, normals, and lighting, but rely on specialized inputs (e.g., HDR captures or spherical harmonics) and typically assume static scenes. This inhibits their practicality for motion-rich, real-world videos.

Recent diffusion models such as RelightVid [8] and Light-A-Video [1] extend the single-image relighting model (i.e., IC-Light [9]) to the video domain. RelightVid [10] modifies the 2D U-Net in IC-Light to a 3D backbone and adds temporal attention layers to enforce frame-to-frame consistency, while Light-A-Video [1] integrates a cross-frame light-attention module into the self-attention blocks of IC-Light and applies a progressive fusion step to suppress flicker and smooth illumination transitions. However, both methods rely on implicit temporal correlations and omit explicit geometric guidance. As such, they suffer from overall loss of lighting fidelity and coherence whenever the scene's geometry changes.

To address these shortcomings, we propose to jointly model lighting and geometry within a unified diffusion framework for controllable video generation. By fusing precise 3D geometry trajectories with detailed illumination cues, our model learns the joint evolution of appearance and motion under dynamic illumination. This approach enables high-fidelity editing of scene illumination, preserving accurate illumination-geometry interactions at every frame.

Specifically, we present IllumiCraft, an end-to-end diffusion architecture that enables controllable video generation, tailored for video relighting, by specifying either: (1) HDR environment maps encoding detailed lighting information; (2) synthetically perturbed video frames with randomized illumination shifts (optionally paired with static background references) to capture appearance variations; and (3) 3D trajectory videos that trace object geometry through space and time. During training, these streams are fused within a DiT-based diffusion model [11, 2], enabling the generation of temporally consistent geometry-aware relit videos. The main contributions of this work are:

- We propose a unified diffusion architecture that jointly incorporates illumination and geometry guidance, enabling high-quality video relighting. It supports text-conditioned and background-conditioned relighting for videos.

- We introduce a high-quality video dataset comprising 20,170 video pairs, featuring paired original videos and synchronized relit videos, HDR maps, and 3D tracking videos. This dataset supports video relighting and serves as a valuable resource for broader controllable video generation tasks.

- We conduct extensive evaluations demonstrating our model's effectiveness against state-of-the-art methods on the video relighting task.

## 2 Related Work

**Diffusion Models.** Building on the success of text-to-image diffusion techniques [12, 13, 14, 15], researchers have fine-tuned diffusion models to tackle a wide range of vision tasks. Interactive image manipulation frameworks like InstructPix2Pix [16], ControlNet [17] and LearnableRegion [18] enable semantic editing conditioned on textual prompts, while geometry-aware extensions such as DreamFusion [19] and DreamPolisher [20] repurpose diffusion priors for text-to-3D generation. In parallel, single-view 3D reconstruction methods [21, 22, 23, 24] leverage score-based priors for 3D reconstruction from a single image. Specialized relighting models [10, 25, 26, 27] have achieved precise per-image illumination adjustments. Recently, IC-Light [9] enforces light-transport independence for image relighting; instead of extending it to video like RelightVid [8] or Light-A-Video [1], we build on the DiT architecture [11] to achieve temporally coherent video relighting.

**Video Diffusion Models and Video Editing**. In recent years, diffusion-based video generation [28, 29, 3, 30, 31, 32] has seen rapid advancements. Building on these foundations, training-free approaches like AnyV2V [33], MotionClone [30], and BroadWay [34] enable prompt-driven inpainting, style transfer, and motion retargeting without additional tuning. For frame-precise coherence, fine-tuning approaches such as ConsistentVideoTune [35] and Tune-A-Video [36] adapt pretrained video diffusion models to user references, delivering seamless object insertion and consistent color grading. Although current video editing approaches [36, 37, 33] achieve impressive performance, they do not explicitly model lighting cues. In contrast, our method embeds both illumination and geometry cues directly into the diffusion model, unlocking the capacity of high-fidelity video relighting.

**Video Relighting.** Recent video relighting techniques have primarily focused on facial videos. SunStage [38] reconstructs facial geometry and reflectance from a rotating outdoor selfie video, using the sun as a natural "light stage" to enable on-device portrait relighting under novel lighting conditions. DifFRelight [39] uses diffusion-based image-to-image translation for free-viewpoint facial performance relighting, conditioning on flat-lit captures and unified lighting controls to produce high-fidelity relit sequences. SwitchLight [40] predicts per-frame normals and shading maps guided by HDR environment maps. More recently, Light-A-Video [1] enhances IC-Light [9] with a consistent light attention module for cross-frame lighting coherence, while RelightVid [8] introduces temporal attention layers into IC-Light to improve temporal consistency. However, they depend exclusively on illumination cues and omit geometric guidance, which hinders their video relighting quality.

## 3 Method

We introduce IllumiCraft, a unified diffusion framework that jointly leverages geometry and illumination cues for controllable video generation. First, in Section 3.1 we describe *IllumiPipe*, our data collection pipeline that constructs a high-quality dataset from real videos, complete with HDR environment maps, 3D tracking video sequences, and synthetically relit foreground clips. Next, Section 3.2 details the core architecture of IllumiCraft, which fuses appearance, geometric, and lighting guidance in a single architecture. Finally, in Section 3.3 and 3.4 we present our training strategy and inference workflow, respectively, highlighting how each component contributes to high-fidelity, controllable video generation.

### 3.1 IllumiPipe

Collecting a paired video dataset with comprehensive annotations is essential for training a robust video generation model capable of supporting high-fidelity video relighting. However, public video datasets rarely include both HDR environment maps and 3D tracking sequences, limiting progress in video relighting and geometry-guided video editing performance. To address this gap, we introduce *IllumiPipe*, an efficient data collection pipeline designed to extract HDR environment maps data, relit video clips, and precise 3D tracking video sequences from real-world videos. Figure 2 illustrates the detailed workflow of *IllumiPipe*.

Specifically, each appearance video $\mathcal{V}_{\text{appr}} \in \mathbb{R}^{T \times H \times W \times 3}$ is accompanied by $6$ types of distinct augmentation data, which together enable joint modeling of geometry and illumination:

$$\mathcal{V}_{\text{appr}} \leftrightarrow \{\mathcal{V}_{\text{rf}}, \mathcal{V}_{\text{bg}}, \mathcal{V}_{\text{hdr}}, \mathcal{V}_{\text{geo}}, \mathcal{V}_{\text{mask}}, \mathcal{C}\},$$

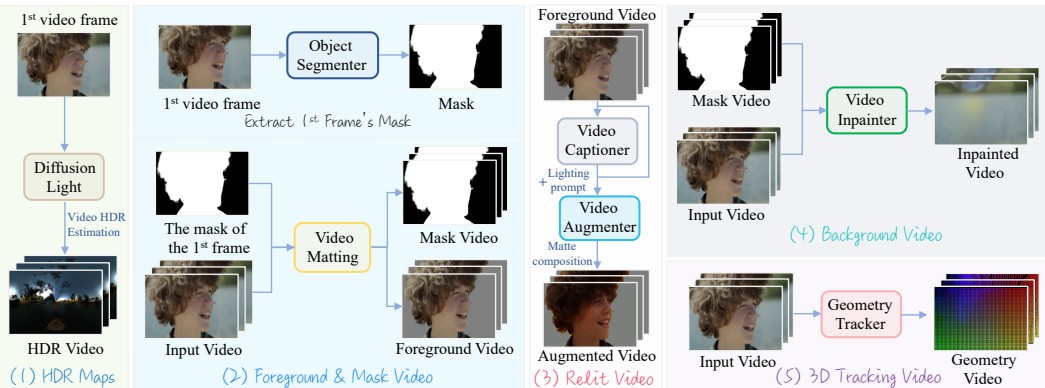

Figure 2: **Data collection mechanism of our proposed IllumiPipe.** For each input video, our proposed IllumiPipe extracts various types of data: (1) HDR maps, (2) foreground video and mask video, (3) relit video, (4) background video, and (5) 3D tracking video. The details and collection process for each data type are described in Section 3.1.

where $\mathcal{V}_{\text{rf}} \in \mathbb{R}^{T \times H \times W \times 3}$ represents the relit foreground video, $\mathcal{V}_{\text{bg}} \in \mathbb{R}^{T \times H \times W \times 3}$ is the background video, $\mathcal{V}_{\text{hdr}} \in \mathbb{R}^{T \times 32 \times 32 \times 3}$ denotes the HDR environment maps, $\mathcal{V}_{\text{geo}} \in \mathbb{R}^{T \times H \times W \times 3}$ represents the 3D tracking video sequences, $\mathcal{V}_{\text{mask}} \in \mathbb{R}^{T \times H \times W \times 3}$ denotes the masks of the foreground video, and $\mathcal{C}$ is the caption describing the appearance video $\mathcal{V}_{\text{appr}}$. Here, $T$, $W$ and $H$ denote the number of frames, frame width and frame height, respectively.

**HDR Environment Maps.** HDR environment maps encode per-direction radiance values that represent both the angular distribution and absolute intensity of scene illumination, enabling physically accurate image-based lighting. We leverage DiffusionLight [41] to extract these HDR maps. However, since DiffusionLight is designed for single-image inputs, applying it independently to each video frame introduces severe temporal inconsistency, where the synthesized chrome ball often varies significantly from frame to frame. To enforce temporal stability, we extract the chrome ball image only from the first frame of each video, and then warp this initial chrome ball onto all subsequent frames, yielding temporally coherent HDR environment maps across the entire sequence.

Specifically, given the chrome ball image from the first frame $\mathcal{I}_c$ (extracted using DiffusionLight [41]), we use Video Depth Anything [42] to obtain depth maps for the video. We then (1) track a sparse set of reliable image points across frames, (2) use their depth values to infer the camera's 3D motion via a constrained affine fit, and (3) apply that motion to warp the reference chrome ball by lifting its pixels to a representative depth, projecting them through the estimated transform and camera intrinsics, and resampling to estimate current frame's chrome ball image. We explain more details about the whole process in the Appendix.

**Relit Videos and Background Videos.** Given a real-world video $\mathcal{V}_{\text{appr}}$, we first apply Grounded SAM-2 [43] to obtain the foreground mask from the first frame. We then feed the appearance video $\mathcal{V}_{\text{appr}}$ together with the first frame's mask into the video object matting model MatAnyone [44] to extract the foreground appearance video $\mathcal{V}_{\text{fg}}$ and the corresponding mask video $\mathcal{V}_{\text{mask}}$.

Next, we generate a relit video $\mathcal{V}_{\text{relit}}$ by applying the video relighting method Light-A-Video [1] to $\mathcal{V}_{\text{fg}}$, conditioned on a diverse set of user-defined lighting prompts. In total, we curated 100 distinct lighting prompts: some are drawn from the official Light-A-Video lighting prompts examples (e.g., *"red and blue neon light"*, *"sunset over sea"*, *"in the forest, magic golden lit"*), while others are generated via GPT-4o mini to cover more challenging or unusual settings (e.g., *"urban jungle glow"*, *"subtle office glow"*). The resulting relit video $\mathcal{V}_{\text{relit}}$ retains the original motion and viewpoint but exhibits the new target illumination. The full list of all 100 lighting prompts is shown in the Appendix. To obtain the background video $\mathcal{V}_{\text{bg}}$, we feed the foreground mask video $\mathcal{V}_{\text{mask}}$ together with the appearance video $\mathcal{V}_{\text{appr}}$ into the video inpainting model DiffEraser [45].

**3D Tracking Videos.** For real-world appearance videos $\mathcal{V}_{\text{appr}}$, where ground-truth geometry is unavailable, we employ SpatialTracker [46] to detect and localize salient 3D interest points directly in 3D space. We initialize a uniform grid of 4,900 points per video to ensure even spatial coverage across the scene. For each consecutive frame pair, SpatialTracker estimates the 3D positions of these points and computes their correspondences using learned spatial matching. The output is a dense and

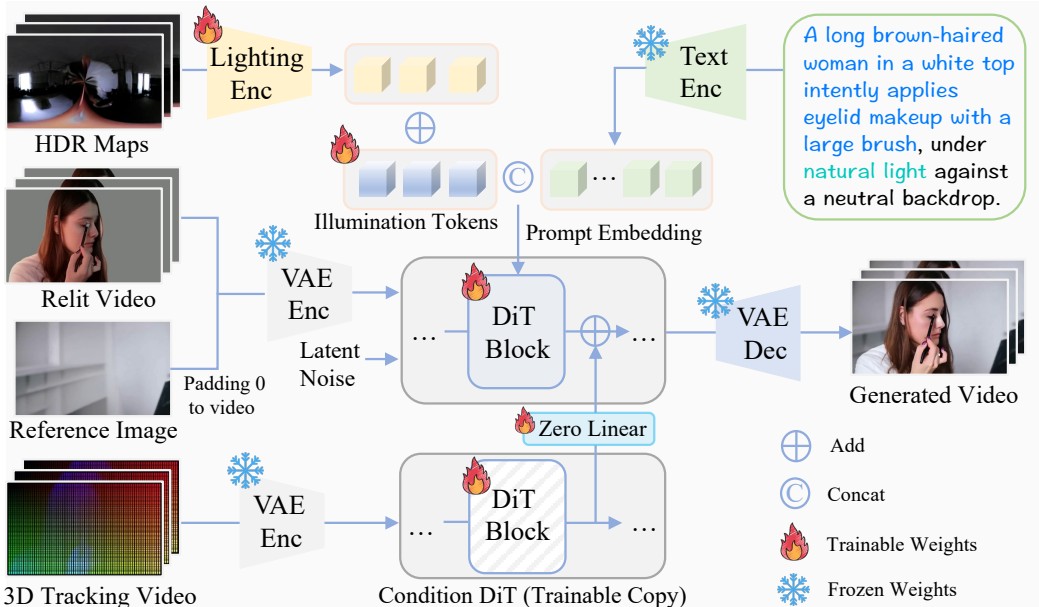

Figure 3: **Famework of IllumiCraft.** It uses HDR maps, relit foreground video, 3D tracking, and an optional background image to jointly model illumination, appearance, and geometry, then generates videos from an illumination-aware text prompt. The figure shows an example with 3 illumination tokens; HDR maps, background images, and 3D tracking videos are all optional during training.

temporally coherent set of 3D point tracks that closely approximates the true motion of the scene, even in unconstrained and dynamic environments.

**Video Captions.** To generate detailed descriptions of each video, we employ CogVLM2-Video-LLaMA3-Chat [47] with the following prompt: *"Carefully watch the video and describe its content, the object's motion, lighting and atmosphere in vivid detail, highlighting the object's movement (e.g., drifting left slowly, darting forward quickly, bouncing up and down, floating upward gently), lighting conditions (e.g., red-and-blue neon, natural sunlight, studio spotlights, sci-fi RGB glow, cyberpunk neon, a sunset over the sea, or magical forest illumination) and the overall atmosphere (e.g., warm, moody, ethereal, cozy, gritty urban, or futuristic haze)."* This prompt guides the model to produce rich captions emphasizing visual content, object motion, lighting, and overall atmosphere.

## 3.2 Model Architecture

We build our framework on top of the pre-trained video generation model Wan2.1 [2], which utilizes a transformer-based video diffusion architecture [11]. By initializing our network with Wan's learned weights, we both leverage its strong video priors and significantly accelerate training. The overall architecture of IllumiCraft is illustrated in Figure 3.

**Latent Feature Extraction.** We first zero-pad the reference image $\mathcal{I}_{\text{ref}} \in \mathbb{R}^{H \times W \times 3}$ (the first frame of the background video $\mathcal{V}_{\text{bg}}$) along the temporal axis to form a reference video $\mathcal{V}_{\text{ref}} \in \mathbb{R}^{T \times H \times W \times 3}$. Note that during training, we randomly zero out the reference image with a 10% probability to enable flexible video editing even when no reference image is provided. Next, we apply the VAE encoder $E_{\text{VAE}}$ to the appearance video $\mathcal{V}_{\text{appr}}$, the relit foreground video $\mathcal{V}_{\text{rf}}$, and the reference video $\mathcal{V}_{\text{ref}}$ to obtain their corresponding latent representations: $z = E_{\text{VAE}}(\mathcal{V}_{\text{appr}}) \in \mathbb{R}^{\frac{T}{4} \times \frac{H}{8} \times \frac{W}{8} \times 16}$, $z_{\text{rf}} = E_{\text{VAE}}(\mathcal{V}_{\text{rf}}) \in \mathbb{R}^{\frac{T}{4} \times \frac{H}{8} \times \frac{W}{8} \times 16}$, and $z_{\text{ref}} = E_{\text{VAE}}(\mathcal{V}_{\text{ref}}) \in \mathbb{R}^{\frac{T}{4} \times \frac{H}{8} \times \frac{W}{8} \times 16}$. We concatenate the relit foreground latent and reference latent along the channel dimension to form the control latent: $z_c = \text{Concat}(z_{\text{rf}}, z_{\text{ref}}) \in \mathbb{R}^{\frac{T}{4} \times \frac{H}{8} \times \frac{W}{8} \times 32}$. We then corrupt the appearance latent $z$ over $t$ diffusion steps to produce $z_t$, concatenate $z_t$ with the control latent $z_c$ along the channel dimension, and feed the result into the diffusion transformer [11, 2] for iterative denoising. Finally, the denoised latent is passed through the VAE decoder $D_{\text{VAE}}$ to reconstruct the output video.

**Inject Illumination Control.** To extract illumination cues from the HDR maps, we first encode the tensor $\mathcal{V}_{\text{hdr}} \in \mathbb{R}^{T \times 32 \times 32 \times 3}$ via the lighting encoder, a compact MLP-Transformer (see Appendix),

producing a feature matrix $\mathcal{X}_{\mathrm{hdr}} \in \mathbb{R}^{N \times D}$. We then introduce the learnable illumination embedding $\mathcal{X} \in \mathbb{R}^{N \times D}$, updated as $\mathcal{X} \leftarrow \mathcal{X} + \mathcal{X}_{\mathrm{hdr}}$. To encourage $\mathcal{X}$ to internalize a robust illumination prior, we randomly zero out $\mathcal{X}_{\mathrm{hdr}}$ with 50% probability during training, simulating the absence of HDR inputs at inference time. To condition the diffusion model, we concatenate $\mathcal{X}$ with the text prompt embedding $\mathcal{P} \in \mathbb{R}^{L \times D}$ to obtain the final prompt embedding $\mathcal{P}' \in \mathbb{R}^{(L+N) \times D}$ (with $N = 3$ in our experiments). At inference, users can steer illumination solely via the text prompt, as $\mathcal{X}$ has learned a standalone representation of lighting priors, eliminating the need for explicit HDR map input.

**Integrate 3D Geometry Guidance.** We extend ControlNet [17] in IllumiCraft by using the 3D tracking video $\mathcal{V}_{\mathrm{geo}} \in \mathbb{R}^{T \times H \times W \times 3}$ as an additional conditioning signal. We encode $\mathcal{V}_{\mathrm{geo}}$ with the VAE encoder $E_{\mathrm{VAE}}$ to produce geometry latents $z_g = E_{\mathrm{VAE}}(\mathcal{V}_{\mathrm{geo}}) \in \mathbb{R}^{\frac{T}{4} \times \frac{H}{8} \times \frac{W}{8} \times 16}$. To inject these signals into DiT, we clone the first 4 blocks of the pretrained 32-block denoising transformer, forming a lightweight "condition DiT": at each cloned block, we pass its output through a zero-initialized linear layer (matching the DiT hidden dimension) and add the result to the corresponding feature map in the main DiT stream, thereby hierarchically fusing geometry information. During fine-tuning, we optimize all DiT blocks under the diffusion denoising loss [31], and we have a 30% possibility of replacing 3D tracking videos with zero tensors to simulate inference without any 3D tracking input.

### 3.3 Training

Building on the DiT backbone [11, 2], we freeze the VAE and CLIP text encoder to retain pretrained priors, optimizing only the DiT blocks and zero-initialized linear layers for stable integration of appearance, geometry, and illumination. Let $\mathcal{E} = \{z_g, z_c, \mathcal{P}'\}$ be the conditional latents and $z_t$ the noisy latent at diffusion step $t$; we then minimize the diffusion loss by predicting the added noise $\epsilon$ as:

$$\min_{\theta} \mathbb{E}_{z \sim E_{\mathrm{VAE}}(x),\, t,\, \epsilon \sim \mathcal{N}(0,1)} \left\| \epsilon \,-\, \epsilon_{\theta}(z_t,\, t,\, \mathcal{E}) \right\|_2^2, \quad \mathcal{E} = \{z_g,\, z_c,\, \mathcal{P}'\}. \tag{1}$$

Here $\epsilon_{\theta}$ is the 3D UNet and $x$ denotes the appearance video $\mathcal{V}_{\mathrm{appr}}$. Conditioning on composite latents $\mathcal{E}$ enables joint reasoning over geometry, appearance, and lighting. Embedding the zero-padded reference frame with text and lighting ensures edits align with the scene and user intent.

### 3.4 Inference

At inference time, users provide a text prompt and an input video, without HDR or geometry to relight the scene, optionally including reference images for video customization. IllumiCraft leverages geometric cues (i.e., 3D tracking videos) during training, and it can generate coherent, illumination-aware videos. We will release the model and curated dataset to support future research.

## 4 Experiments

### 4.1 Baselines and Evaluation Metrics

Our work addresses the task of video relighting, benchmarking our approach against 4 state-of-the-art approaches: **IC-Light** [9], adapted to video by processing each frame independently; **IC-Light + AnyV2V** [33], where IC-Light relights only the first frame, and AnyV2V then propagates those changes to later frames while preserving the original content; **RelightVid** [8], which natively supports the first 16 frames (we therefore report results on both the full 49-frame video sequence and on its first 16 frames); and **Light-A-Video** [1], using Wan2.1 1.3B [2] as its backbone (the same base model used in our method). All evaluation results are based on each baseline's publicly released code.

We report 5 metrics for evaluation. The visual quality of the generated videos is evaluated by computing the FVD [48], LPIPS [49] and PSNR [50] scores against the results of the existing methods. Text alignment is quantified as the mean CLIP [51] cosine similarity score between each individual frame and the text prompt. Temporal consistency, on the other hand, is measured as the average CLIP cosine similarity score across every pair of consecutive frames.

### 4.2 Implementation Details

We collect 20,170 high-quality, free-to-use videos from Pexels (https://www.pexels.com/) for training. Wan2.1 1.3B [2] serves as the DiT backbone. Models are trained for 3,000 iterations on

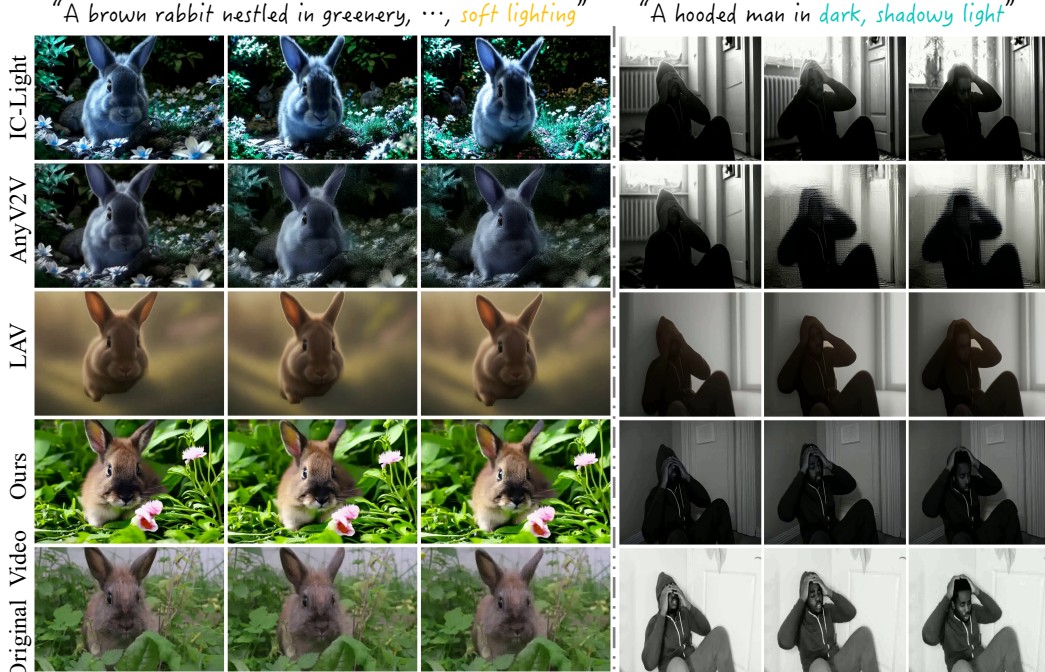

Figure 4: **Visual results under the text-conditioned setting.** We compare IC-Light [9], AnyV2V [33], Light-A-Video [8] (abbreviated LAV in the figure), and our proposed method, IllumiCraft.

Table 1: Quantitative comparison of text-conditioned video relighting with existing methods.

| Method | FVD ($\downarrow$) | LPIPS ($\downarrow$) | PSNR ($\uparrow$) | Text Alignment ($\uparrow$) | Temporal Consistency ($\uparrow$) |
|---|---|---|---|---|---|
| IC-Light [9] | 4914.83 | 0.7330 | 8.55 | 0.3091 | 0.9508 |
| AnyV2V [33] + IC-Light | 3857.09 | 0.6979 | 11.12 | 0.2781 | 0.9808 |
| Light-A-Video [1] | 3946.71 | 0.6754 | 11.71 | 0.3020 | 0.9910 |
| IllumiCraft | 2186.40 | 0.5623 | 12.03 | 0.3342 | 0.9948 |

4×A6000 GPUs with a batch size of 2 and gradient accumulation over 4 steps, taking approximately 90 hours. Videos are center-cropped and resized to 720×480 (width×height) resolution with 49 frames. We use the AdamW optimizer [52] with a learning rate of 4e-5, weight decay of 0.001, and 100 warmup steps. The learning rate follows a cosine schedule with restarts. For evaluation, we collect an additional 50 high-quality Pexels videos and show the results on this set.

### 4.3 Comparison with State-of-the-art Methods

**Text-Conditioned Video Relighting.** In this setting, only a foreground video and a text prompt are provided for relighting. As shown in Table 1, our method significantly outperforms prior work, achieving the lowest FVD and improved perceptual quality across all metrics. Compared to baselines like IC-Light [9], AnyV2V [33], and Light-A-Video [1], our approach yields sharper visuals, better alignment with text descriptions, and higher temporal stability. For instance, we observe a 43% reduction in FVD compared to the strongest baseline. Qualitative comparisons in Figure 4 further illustrate the differences: under prompts like "soft lighting" for a rabbit or "dark, shadowy light" for a man, IC-Light produces overly smoothed fur, AnyV2V introduces color distortions, and Light-A-Video blurs fine details and dampens contrast. In contrast, IllumiCraft preserves fine textures, captures lighting nuances, ensures prompt relevance, and produces flicker-free, coherent videos.

**Background-Conditioned Video Relighting.** We also evaluate on background-conditioned video relighting, where both the input foreground and background are given. Table 2 reports results across multiple baselines, including IC-Light [9], AnyV2V [33] + IC-Light, Light-A-Video [1], and RelightVid [8]. Our method consistently achieves superior performance in both short (16-frame) and long (49-frame) sequences. For example, on 49-frame inputs, our method cuts FVD by 37% compared to Light-A-Video, while improving perceptual similarity, alignment with prompts, and temporal consistency. On 16-frame sequences, we further improve fidelity and detail preservation,

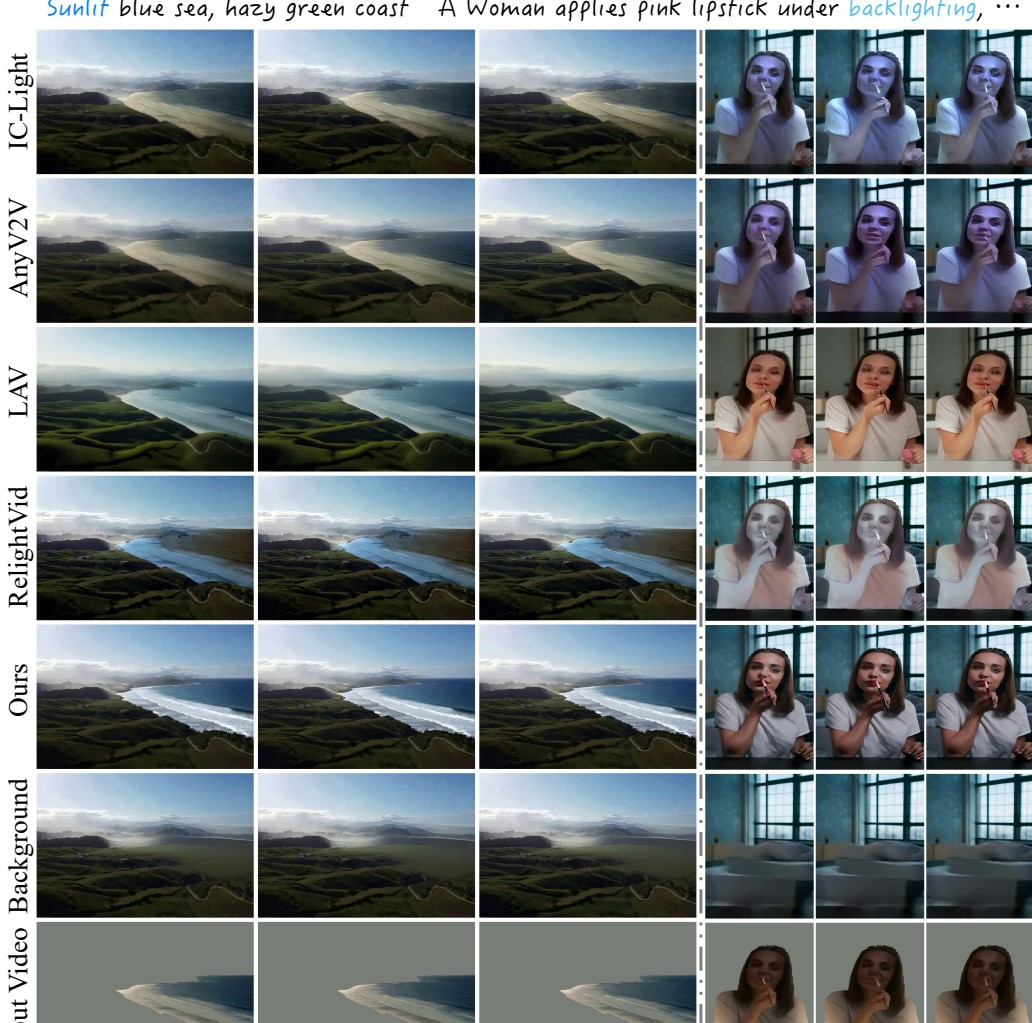

Figure 5: **Visual results under the background-conditioned setting.** We compare IC-Light [9], AnyV2V [33], Light-A-Video [8], RelightVid [8] and our proposed method, IllumiCraft.

Table 2: Quantitative comparison of background-conditioned video relighting with existing methods. * denotes results evaluated with the first 16 frames.

| Method | FVD (↓) | LPIPS (↓) | PSNR (↑) | Text Alignment (↑) | Temporal Consistency (↑) |
|---|---|---|---|---|---|
| IC-Light [9] | 2175.97 | 0.3049 | 17.20 | 0.3037 | 0.9795 |
| AnyV2V [33] + IC-Light | 1901.41 | 0.3447 | 17.98 | 0.3021 | 0.9854 |
| Light-A-Video [1] | 1704.63 | 0.3834 | 15.64 | 0.3266 | 0.9912 |
| RelightVid* [8] | 1492.18 | 0.2989 | 17.19 | 0.3055 | 0.9858 |
| IllumiCraft* | 1011.08 | 0.2232 | 19.78 | 0.3283 | 0.9932 |
| IllumiCraft | 1072.38 | 0.2592 | 19.44 | 0.3292 | 0.9945 |

outperforming RelightVid [8] in every metric. As shown in Figure 5, when relighting scenes like sunlit blue sea and a backlit woman, competing methods struggle with color shifts, exposure mismatches, or excessive smoothing. Instead, IllumiCraft delivers prompt-faithful lighting, clear subject-background separation and highly realistic textures, yielding the most photorealistic and coherent outputs.

## 4.4 Ablation Studies

**Impact of Illumination and Geometry Guidance.** In Table 3, we compare using illumination-only (I) versus illumination combined with geometry guidance (I+G) during training. Incorporating geometry leads to consistent improvements across metrics, including a ~18% reduction in FVD and

Table 3: Ablation study of adopting different guidance. "I" and "G" denote the illumination and geometry guidance respectively, used during training.

| Guidance | FVD (↓) | LPIPS (↓) | PSNR (↑) | Text Alignment (↑) | Temporal Consistency (↑) |
|---|---|---|---|---|---|
| G | 1157.23 | 0.2654 | 18.89 | 0.2781 | 0.9932 |
| I | 1305.45 | 0.2816 | 18.28 | 0.3211 | 0.9864 |
| I + G | 1072.38 | 0.2592 | 19.44 | 0.3292 | 0.9945 |

Table 4: Effect of dropping $\mathcal{X}_{hdr}$ (text-only).

| Possibility | FVD (↓) | TA (↑) | TC (↑) |
|---|---|---|---|
| 40% | 2172.35 | 0.3325 | 0.9942 |
| 50% | 2186.40 | 0.3342 | 0.9948 |
| 60% | 2123.23 | 0.3312 | 0.9932 |
| 70% | 2138.35 | 0.3301 | 0.9923 |

Table 5: Effect of dropping $\mathcal{X}_{hdr}$ (background).

| Possibility | FVD (↓) | TA (↑) | TC (↑) |
|---|---|---|---|
| 40% | 1048.24 | 0.3265 | 0.9926 |
| 50% | 1072.38 | 0.3292 | 0.9945 |
| 60% | 1065.21 | 0.3277 | 0.9937 |
| 70% | 1051.14 | 0.3228 | 0.9910 |

better perceptual quality, alignment, and temporal consistency. These results suggest that geometry offers critical spatial context that complements illumination cues, helping the model better understand surface structure and light interaction. As a result, the relit videos are not only more photorealistic, but also maintain better alignment with the input prompt and show fewer temporal artifacts.

**Effect of Dropping** $\mathcal{X}_{hdr}$**.** We study the effect of partially dropping $\mathcal{X}_{hdr}$ by varying the drop rate from 40% to 70% and analyzing its influence on FVD, text alignment (TA), and temporal consistency (TC) for both text- and background-conditioned relighting (Tables 4 and 5). A 50% drop rate consistently yields the best trade-off between visual fidelity and alignment with text prompts. This shows that a moderate omission rate preserves essential high-dynamic-range cues while improving generalization and temporal consistency.

**Inference Cost.** All experiments were conducted on an NVIDIA A6000 GPU. For a 49-frame video at 720×480, IC-Light [9] takes 228.31 seconds via frame-wise relighting (about 4.66 seconds/frame), AnyV2V [33] requires 1033.21 seconds in total (649.10 seconds for DDIM inversion [53] + 384.11 seconds for frame editing), and Light-A-Video [1] finishes in 645.53 seconds using its progressive light-injection pipeline. In contrast, IllumiCraft completes inference in just 105.21 seconds per sample owing to its efficient spatio-temporal latent encoding. When relighting a 16-frame video at 720×480, RelightVid takes 23.24 seconds, whereas our model requires only 22.12 seconds, faster while achiving higher visual fidelity. Additional experiments are included in the Appendix.

## 4.5 Discussion

While geometry guidance offers powerful control over 3D scenes, and even enables broader controllable video generation tasks, our work tackles the more demanding challenge of video relighting. In video relighting, each frame's illumination must remain temporally consistent while adapting to the scene's evolving geometry. To achieve this, IllumiCraft is trained with both 3D-geometry cues and illumination cues, enforcing stable lighting across all frames. We build on DiffusionLight [41] to obtain HDR environment maps as our illumination guidance, thus the performance of DiffusionLight can influece the lighting quality of relit videos. To further reduce artifacts like misplaced shadows, we can extend the current collected video dataset under a wider variety of lighting conditions.

## 5 Conclusion

In this work, we present IllumiCraft, a unified diffusion framework that integrates geometry and illumination guidance for controllable video generation. To support high-quality and diverse relighting control, we curate a large-scale dataset of 20,170 video pairs, enriched with HDR maps and 3D tracking sequences. Extensive experiments demonstrate the effectiveness of the proposed method.

**Limitations and Broader Impact.** While our approach enables effective video relighting in diverse scenes, its fidelity depends on the generative prior of the base model. In cases where this prior lacks accurate shading cues or high-frequency details, the output may exhibit artifacts such as texture blurring. In addition, by enhancing illumination realism and temporal coherence, our method may inadvertently increase the credibility of manipulated videos, raising ethical concerns around deepfakes. We encourage future work on safeguards and detection techniques to mitigate potential misuse.

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

# 6  Appendix

## A  Overview

In the Appendix, we provide the following content:

   (a) Details of HDR environment map transformation in Section B.
   (b) Details of the presented lighting encoder in Section C.
   (c) Additional experimental results in Section D.
   (d) Additional visualization results in Section E.

## B  HDR Map Transformation

Let $\mathcal{I}_c$ denote the first (base) video frame, from which we extract a reference chrome ball image using DiffusionLight [41]. Then we relabel $\mathcal{I}_c$ as $\mathcal{I}_0$ and denote each subsequent frame by $\mathcal{I}_t$ for $t \in [1, T]$, where $T$ is the total number of frames (49 in our experiments). We use Video-Depth-Anything [42] to provide per-frame depth maps $D_t(u, v)$. Our goal is to synthesize the appearance of the chrome ball in each $\mathcal{I}_t$ by estimating the camera's 3D motion and warping $\mathcal{I}_0$ accordingly.

### B.1  Sparse Feature Tracking

From the base frame $\mathcal{I}_0$, we detect up to $N = 200$ reliable 2D corners $\boldsymbol{p}_i^0 = (u_i^0, v_i^0), t \in [1, N]$ using the OpenCV Shi-Tomasi detector and track them into the current frame $\mathcal{I}_t$ using the OpenCV Lucas-Kanade optical flow, which produces $\boldsymbol{p}_i^t = (u_i^t, v_i^t)$; points with failed tracks or missing depth are discarded, leaving a robust set of correspondences for motion estimation.

### B.2  3D Motion Estimation via Constrained Affine Fit

For each correspondence $(\mathbf{p}_i^0, \mathbf{p}_i^t)$, we first read the depths $z_i^0 = D_0(u_i^0, v_i^0)$ and $z_i^t = D_t(u_i^t, v_i^t)$ from the base and current depth maps, then lift them to the 3D points of normalized camera by $\mathbf{X}_i^0 = z_i^0 K^{-1}[u_i^0, v_i^0, 1]^\top$ and $\mathbf{X}_i^t = z_i^t K^{-1}[u_i^t, v_i^t, 1]^\top$, where $K$ is the intrinsic matrix. We estimate the rigid affine transform $(R, t)$ by minimizing $\sum_i \|R\mathbf{X}_i^0 + t - \mathbf{X}_i^t\|^2$ subject to $R^\top R = I$ and $\det R = 1$ using the OpenCV `estimateAffine3D`. To enforce temporal smoothness, we dampen the raw 3×4 transform $M_{3d} = [R \mid t]$ toward the 3×4 identity affine matrix $\hat{I}$ via $M_{3d} \leftarrow \hat{I} + \alpha (M_{3d} - \hat{I})$ with $\alpha = 0.05$, and finally re-orthogonalize $R$ via SVD while clamping both rotation angle and translation magnitude.

### B.3  Warping the Reference Chrome Ball

For each pixel $(x_0, y_0)$ in the reference chrome ball image $\mathcal{I}_0$ of size $(w, h)$, we compute the mean depth $d_{\text{avg}} = \frac{1}{WH} \sum_{u=0}^{W-1} \sum_{v=0}^{H-1} D^0(u, v)$, map $(x_0, y_0)$ into video-frame coordinates by $x_v = \frac{W}{w} x_0$ and $y_v = \frac{H}{h} y_0$, lift it to 3D via $\mathbf{X}_0 = d_{\text{avg}} K^{-1}[x_v, y_v, 1]^\top$, apply the affine transform $\mathbf{X}_0' = R\mathbf{X}_0 + t$, project back via $[u', v', 1]^\top \propto K \mathbf{X}_0'$, recover warped coordinates $x_0' = \frac{u'}{W/w}$ and $y_0' = \frac{v'}{H/h}$, and finally use the OpenCV `remap` to sample $\mathcal{I}_0$ at $(x_0', y_0')$, producing the warped chrome ball in the current frame.

### B.4  Discussion

Our method tracks a few key points on the chrome ball in 3D using depth maps to directly recover camera motion, avoiding the pitfalls of 2D model fitting. We reduce jitter and ensure smooth results by gently smoothing each new motion estimate and capping its maximum change. Since the chrome ball is nearly spherical, using its average depth introduces only negligible error. This fully automatic pipeline produces accurate, stable warps with minimal manual intervention.

**Robustness to HDR/3D Inputs.** During training, we utilize high-quality, aligned HDR maps and 3D geometry data to help the model learn robust priors for spatial structure and lighting. However,

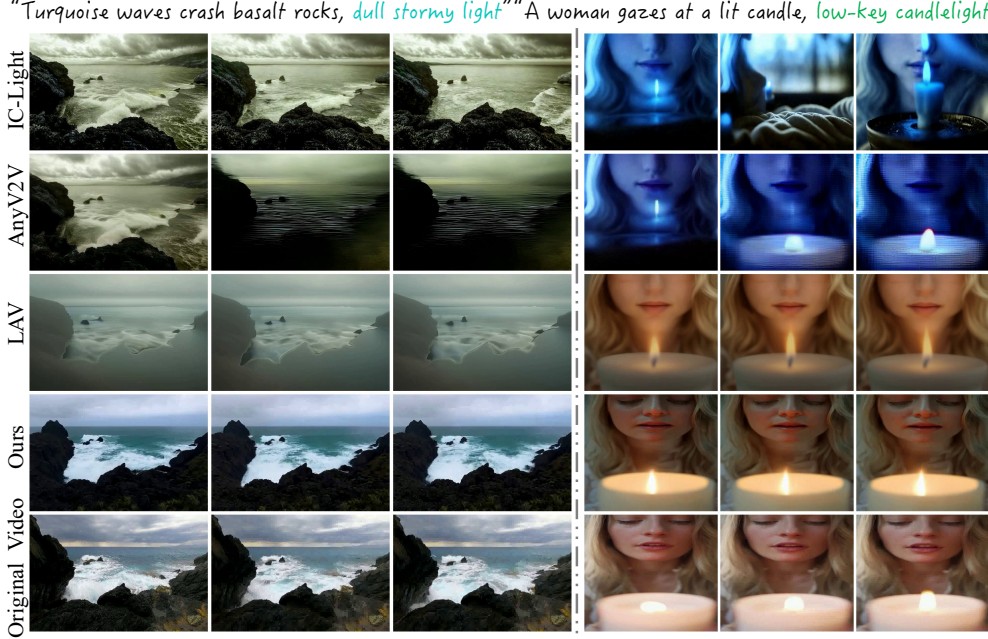

Figure 6: **Visual results under the text-conditioned setting.** We compare IC-Light [9], AnyV2V [33], Light-A-Video [8] (abbreviated LAV in the figure), and our proposed method, IllumiCraft.

these auxiliary signals are not required at inference. For video relighting evaluation, our model only takes the input video, a text prompt, and an optional background image as inputs. It generates relit results solely based on priors learned during training, without relying on HDR maps or 3D tracking at test time. This design ensures that the model is inherently robust to any potential HDR or 3D tracking failure at inference, including under fast motion.

**Collected Lighting Prompts.** To generate truly diverse relit videos, we curated 100 unique lighting prompts (see Table 11 and Table 12). These prompts span both everyday and fantastical illumination scenarios, ranging from simple indoor scenes to dramatic, otherworldly effects. This variety ensures that our model delivers outputs that are both quantitatively robust and qualitatively rich.

## C  Lighting Encoder

The lighting encoder first reshapes the input HDR video tensor $\mathcal{V}_{\text{hdr}} \in \mathbb{R}^{T \times 32 \times 32 \times 3}$ (with $T = 49$) into $X_1 \in \mathbb{R}^{49 \times 3072}$, then applies a four-layer MLP, each Linear layer followed by LeakyReLU, with dimensions $3072 \rightarrow 4096 \rightarrow 4096 \rightarrow 4096 \rightarrow 4608$ (where $4608 = 3 \times 1536$, corresponding to 3 illumination tokens) to produce $Y_1 \in \mathbb{R}^{49 \times 4608}$. This is reshaped and permuted into $Y_2 \in \mathbb{R}^{3 \times 49 \times 1536}$, processed by a single-layer TransformerEncoder ($d_{\text{model}} = 1536$, $n_{\text{head}} = 8$, $\dim_{\text{ff}} = 2048$) yielding $Y_3 \in \mathbb{R}^{3 \times 49 \times 1536}$, and then passed through a depth-wise Conv1d followed by LeakyReLU and squeeze, to collapse the temporal axis into the final output $Z \in \mathbb{R}^{3 \times 1536}$.

## D  Additional Experimental Results

### D.1  Comparison with Existing Methods

**Text-Conditioned Video Relighting.** Figure 6 qualitatively benchmarks four video relighting methods on two different illumination conditions, dull stormy light and a low-key candle light, showing side-by-side comparisons of IC-Light [9], AnyV2V [33], Light-A-Video [1] and IllumiCraft. In the coastal scene, LAV yields overly smooth, desaturated outputs, AnyV2V introduces temporal jitters and erratic color shifts. IC-Light also causes color shifts and cannot preserve the fine details in the original video frames. In contrast, IllumiCraft preserves the original structure, faithfully renders

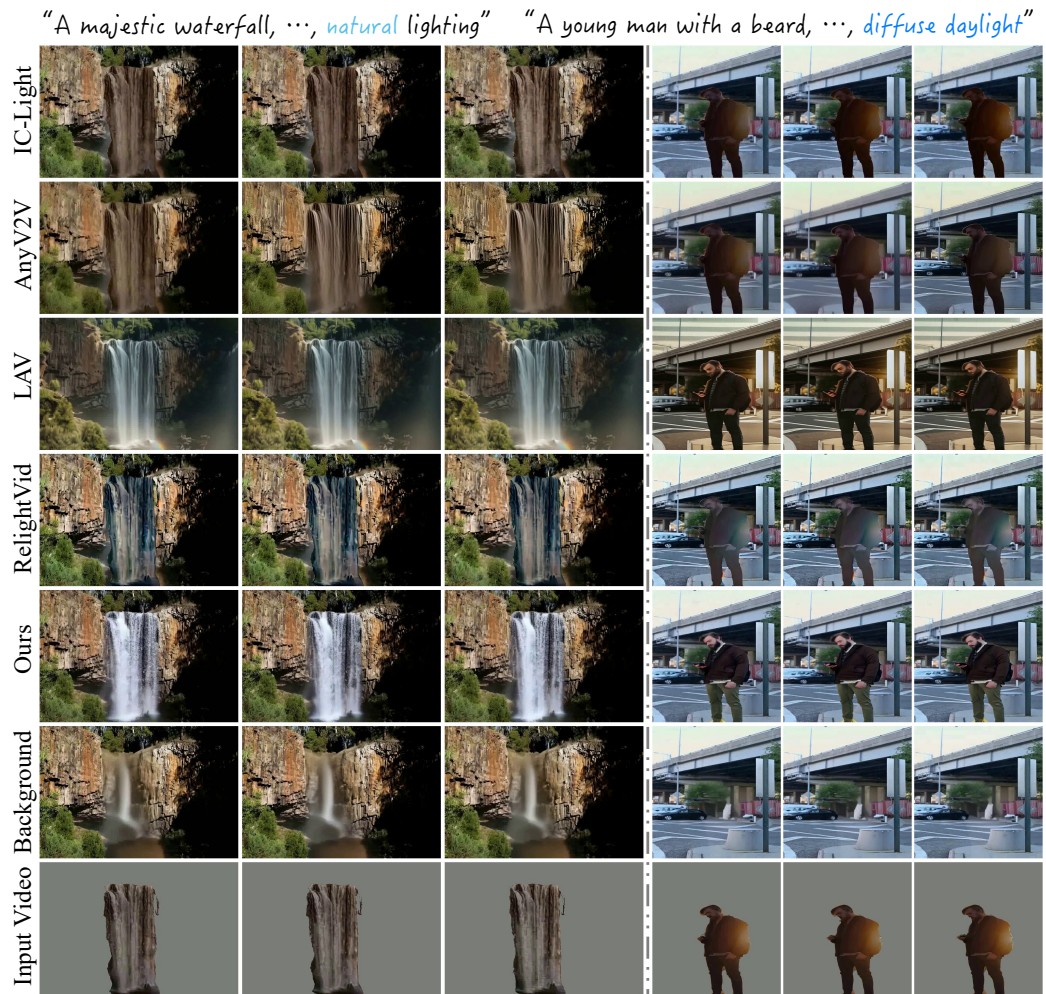

Figure 7: **Visual results under the background-conditioned setting.** We compare IC-Light [9], AnyV2V [33], Light-A-Video [8], RelightVid [8] and our proposed method, IllumiCraft.

Table 6: Impact of dropping 3D tracking videos (text-only).

| Possibility | FVD ($\downarrow$) | TA ($\uparrow$) | TC ($\uparrow$) |
|---|---|---|---|
| 10% | 2285.32 | 0.3303 | 0.9893 |
| 20% | 2251.21 | 0.3332 | 0.9939 |
| 30% | 2186.40 | 0.3342 | 0.9948 |
| 40% | 2234.35 | 0.3325 | 0.9915 |

Table 7: Effect of dropping 3D tracking videos (background).

| Possibility | FVD ($\downarrow$) | TA ($\uparrow$) | TC ($\uparrow$) |
|---|---|---|---|
| 10% | 1154.32 | 0.3231 | 0.9902 |
| 20% | 1102.21 | 0.3273 | 0.9935 |
| 30% | 1072.38 | 0.3292 | 0.9945 |
| 40% | 1098.35 | 0.3278 | 0.9937 |

prompt-specific cues (e.g., turquoise waves, intimate candlelight glow), and maintains temporal stability without artifacts. This demonstrates superior fidelity and consistency over all baselines.

**Background-Conditioned Video Relighting.** As shown in Figure 7, we compare two relighting scenarios, a majestic waterfall under natural lighting (left) and a bearded man under diffuse daylight (right), over four baselines (IC Light [9], AnyV2V [33], Light-A-Video [1] and RelightVid [8]) and our method. RelightVid introduces banding and creates unnatural illumination on the waterfall. IC Light and AnyV2V preserve the overall brightness, but blur fine details such as droplets, hair, and clothing. Light-A-Video desaturates tones, oversmooths the water spray, and alters the portrait background, causing artifacts. In contrast, our method follows each prompt precisely, achieving a high-fidelity waterfall and sharp rock edges with rock-solid frame-to-frame consistency, enhancing detail preservation and temporal coherence in both scenarios.

"An ancient stone castle on an island, ⋯, bright chalky spotlight in misty blue haze"

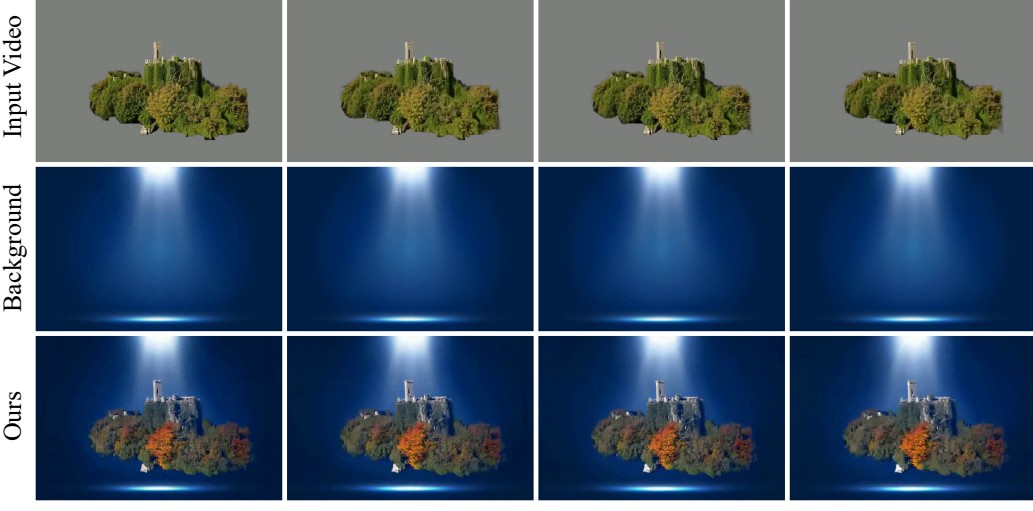

"branch of an apple tree in full bloom, ⋯, towering LED floodlight, stark snowy illumination"

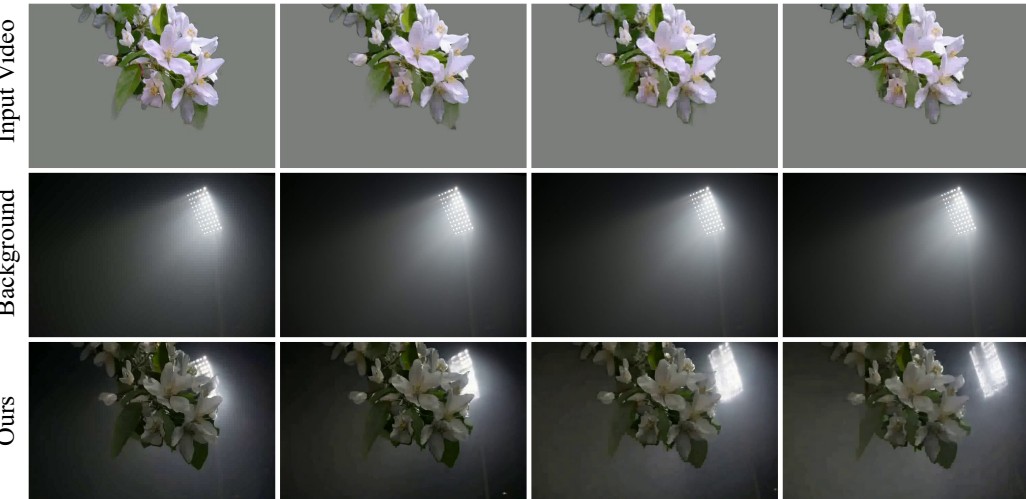

Figure 8: **Failure cases of IllumiCraft.** We show the failure cases generated by our method.

Table 8: Effect of dropping the reference image (text-only).

| Possibility | FVD (↓) | TA (↑) | TC (↑) |
|---|---|---|---|
| 5% | 2232.46 | 0.3331 | 0.9939 |
| 10% | 2186.40 | 0.3342 | 0.9948 |
| 20% | 2175.23 | 0.3341 | 0.9943 |
| 30% | 2158.32 | 0.3338 | 0.9941 |

Table 9: Effect of dropping the reference image (background).

| Possibility | FVD (↓) | TA (↑) | TC (↑) |
|---|---|---|---|
| 5% | 1065.83 | 0.3284 | 0.9941 |
| 10% | 1072.38 | 0.3292 | 0.9945 |
| 20% | 1105.28 | 0.3275 | 0.9928 |
| 30% | 1127.32 | 0.3269 | 0.9925 |

## D.2 Ablation Study

**Impact of Dropping 3D Tracking Videos.** We evaluate the impact of randomly dropping 3D tracking videos during training under both text-conditioned (Table 6) and background-conditioned (Table 7) settings. A 30% drop rate offers the best balance across visual quality, text alignment, and frame consistency. In the text-conditioned scenario, it reduces FVD, increases the text alignment score to 0.3342, and improves the temporal coherence score to 0.9948. In the background-conditioned setting, the same drop rate also lowers FVD and boosts text alignment to 0.3292. These results suggest that a 30% drop rate consistently yields optimal performance across all key metrics.

Table 10: Ablation study on per-frame HDRs and our HDR warping method.

| HDR | FVD ($\downarrow$) | LPIPS ($\downarrow$) | PSNR ($\uparrow$) | Text Alignment ($\uparrow$) | Temporal Consistency ($\uparrow$) |
|---|---|---|---|---|---|
| Per-frame | 1267.23 | 0.2689 | 17.95 | 0.3102 | 0.9901 |
| HDR Warping (Ours) | 1072.38 | 0.2592 | 19.44 | 0.3292 | 0.9945 |

**Effect of Dropping Reference Image.** As shown in Tables 8 and 9, a 10% possibility of dropping reference images during training achieves the best overall trade-off in both text-only and background-only settings. In the text-only condition, while the 10% drop rate does not result in the lowest FVD, it delivers the highest text alignment (0.334) and the highest temporal coherence (0.995), making it the most balanced choice. In the background-only scenario, the same 10% drop rate lowers FVD to 1072.38, raises text alignment to 0.3292, and maintains coherence at 0.9945. Overall, a 10% drop rate maximizes visual realism, alignment, and frame consistency across both settings.

**Effect of HDR Warping.** To evaluate the effectiveness of HDR warping in training, we compare our HDR warping strategy with per-frame HDRs estimated by DiffusionLight [41] under the background-conditioned setting. As shown in Table 10, our HDR warping consistently achieves better results across all metrics, including lower FVD and LPIPS, higher PSNR and text alignment, and improved temporal consistency. These findings confirm that incorporating temporally consistent HDR warping during training provides a significant advantage for video relighting.

### D.3 Failure Cases

As shown in Figure 8, in the top example (an ancient stone castle illuminated by a bright chalky spotlight in misty blue haze), our relighting sometimes shifts and misaligns the lower foliage, resulting in oversaturated greens. In the bottom example (a flowering apple branch moving in front of a towering LED floodlight), when the branch crosses the illuminated region, parts of the floodlight are occluded and mistakenly treated as foreground, causing unwanted changes in the floodlight's appearance. To address these issues, we plan to expand our curated dataset to include more scenes with dynamic occlusions and strong directional lighting.

## E   Additional Visualization Results

Figure 9, 10, 11, 12, 13 and 14 show additional relit videos generated by IllumiCraft. Across a wide range of custom backgrounds, including varied spotlight arrangements, colored backdrops, and complex scene contents, IllumiCraft adapts seamlessly to diverse illumination scenarios, producing smooth shading transitions, consistent specular highlights, and accurate shadows. These examples demonstrate its versatility in handling challenging lighting conditions while faithfully preserving original background details, resulting in high-quality relit videos without visible artifacts.

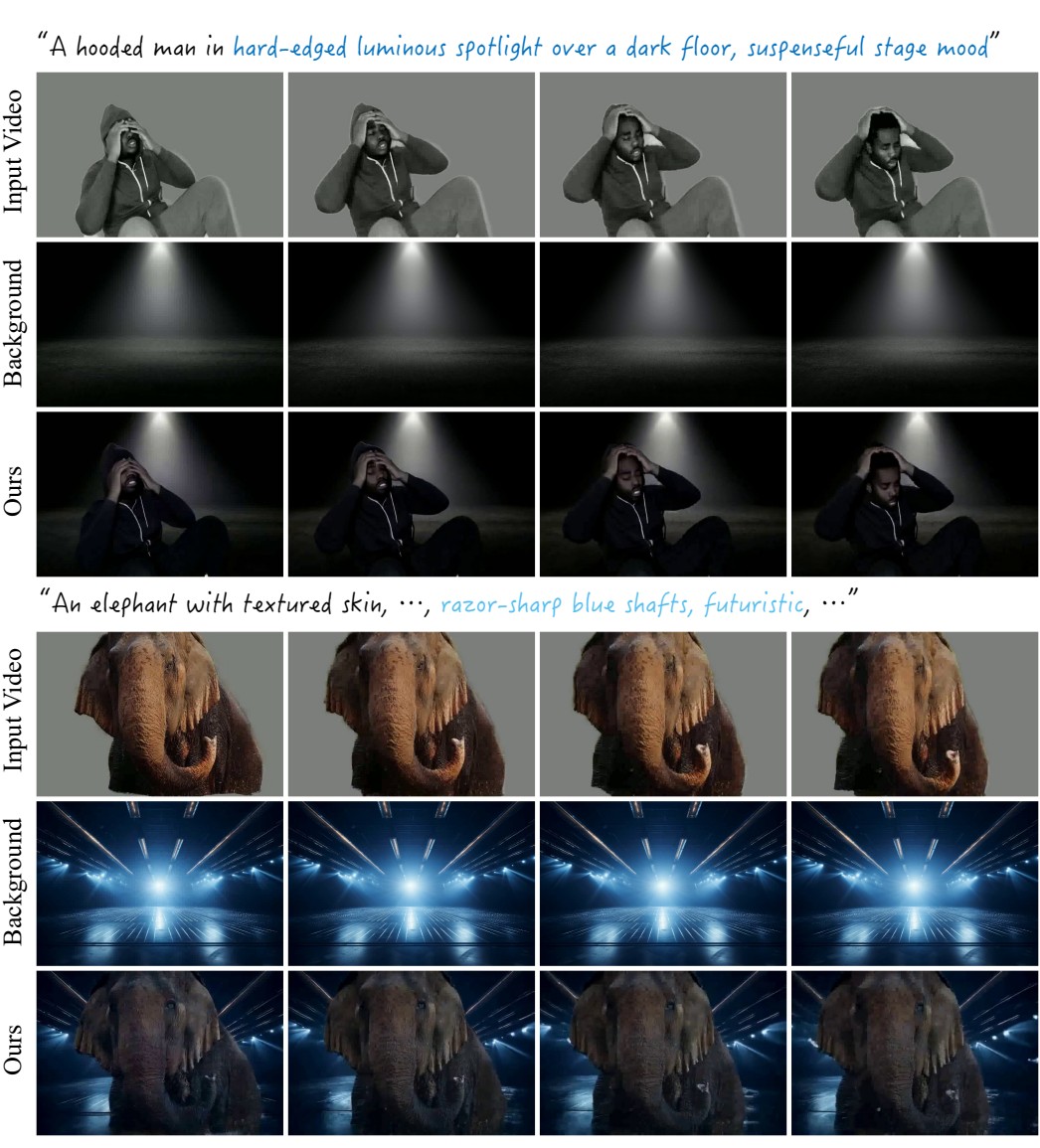

Figure 9: **Visual results of IllumiCraft.** Our method produces high-fidelity, prompt-aligned videos that adapt to diverse lighting conditions, including dramatic spotlight effects.

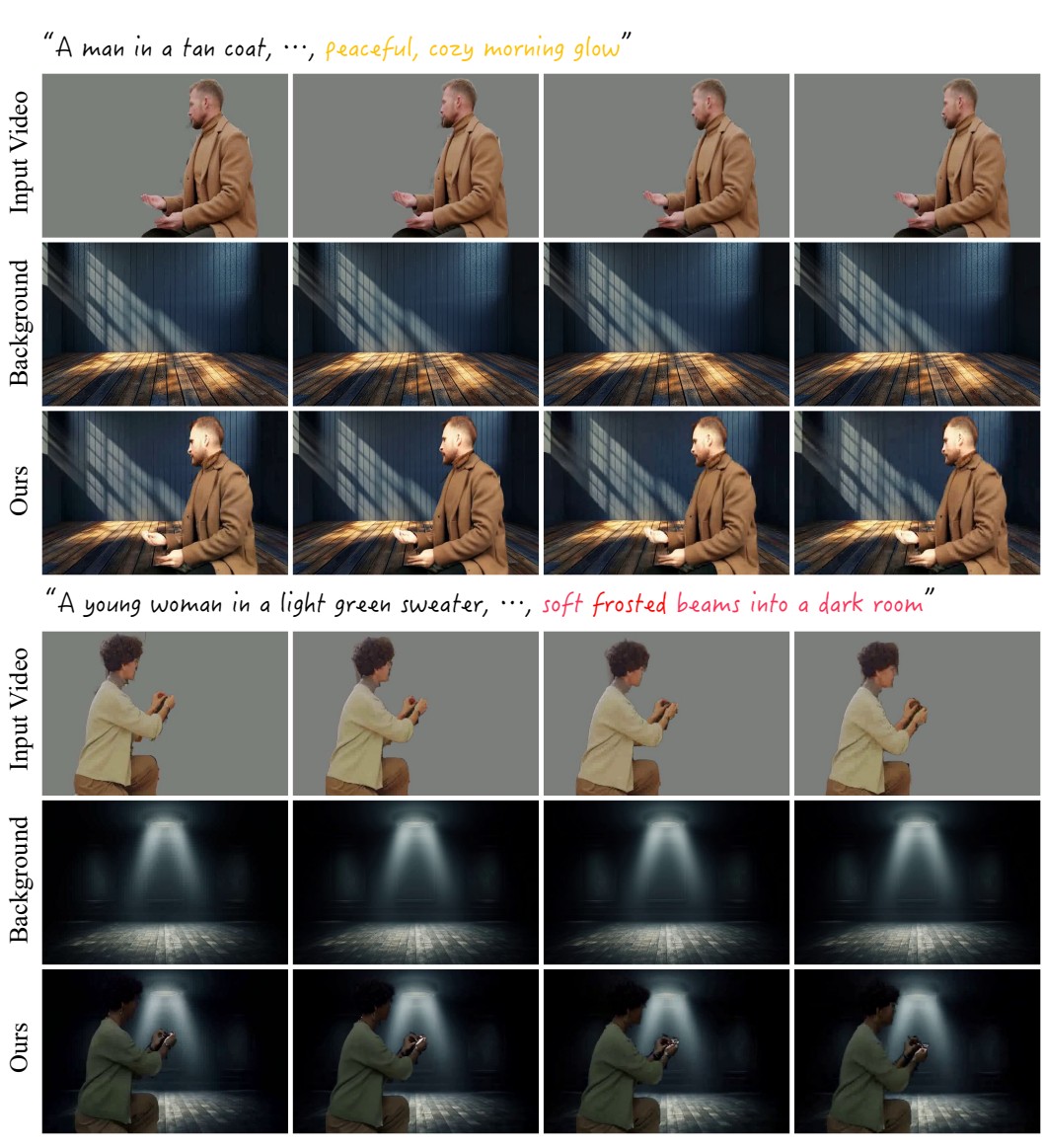

Figure 10: **Visual results of IllumiCraft.** Our method produces high-fidelity, prompt-aligned videos that adapt to diverse lighting conditions, including dramatic spotlight effects.

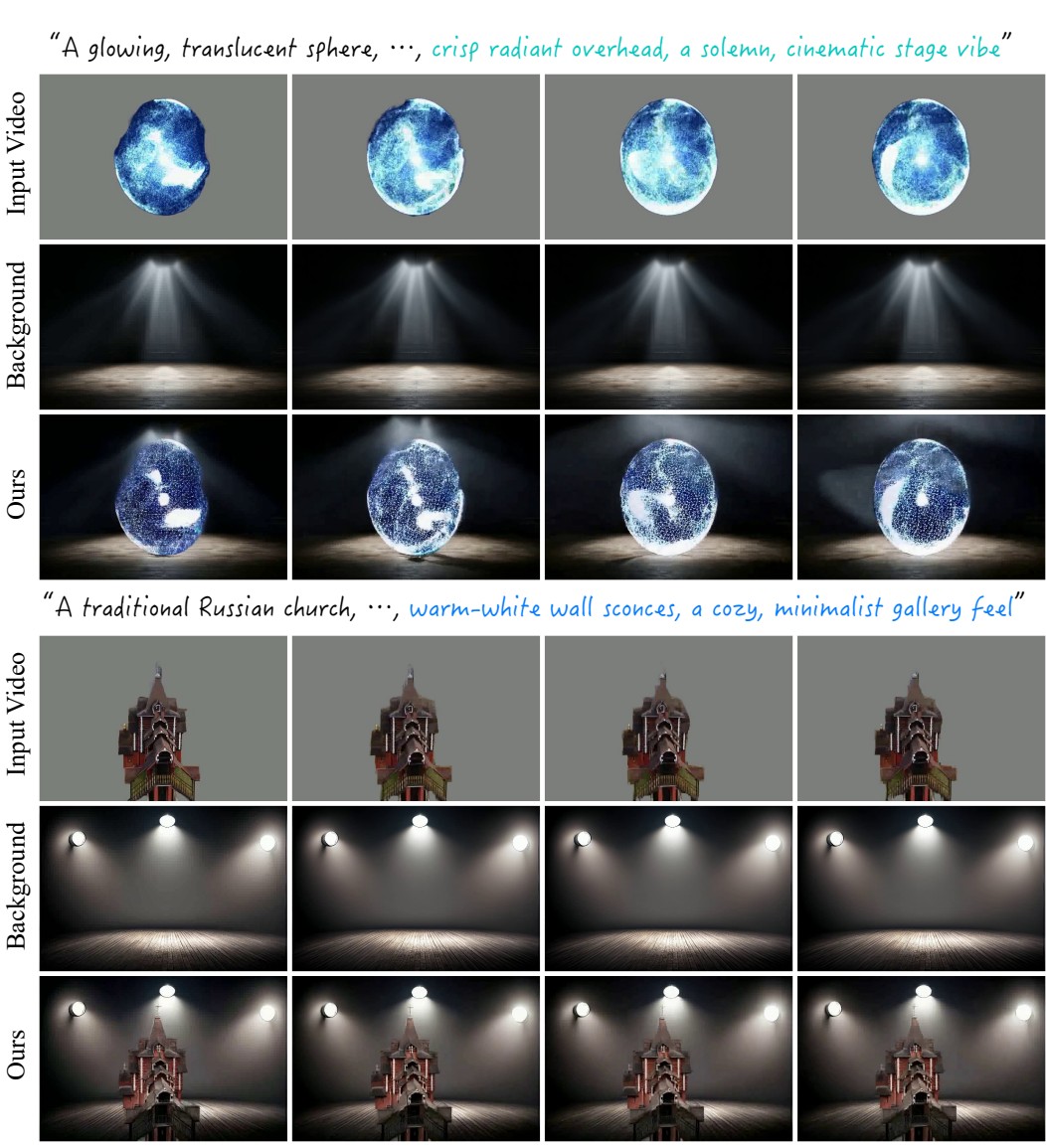

Figure 11: **Visual results of IllumiCraft.** Our method produces high-fidelity, prompt-aligned videos that adapt to diverse lighting conditions, including dramatic spotlight effects.

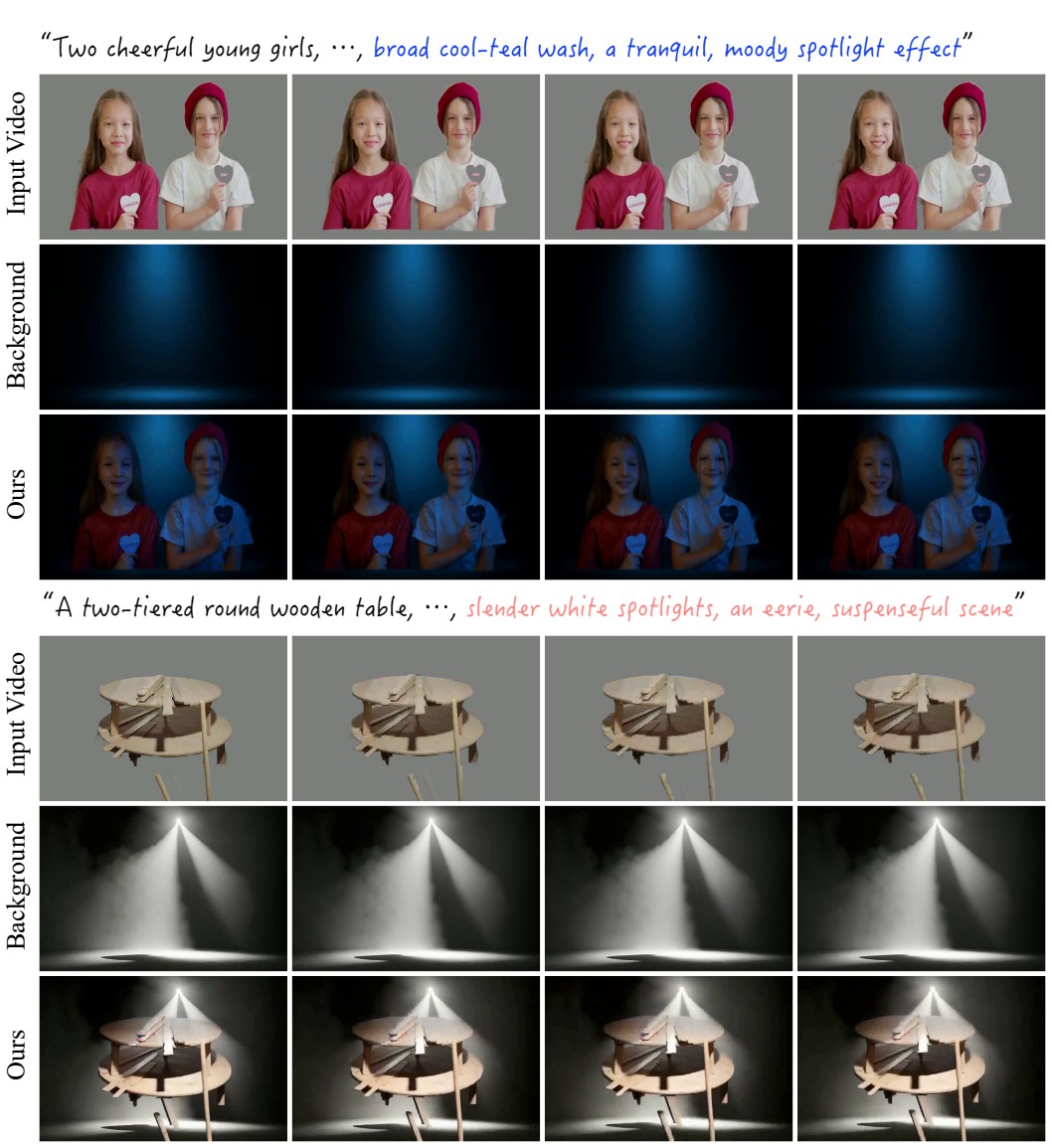

Figure 12: **Visual results of IllumiCraft.** Our method produces high-fidelity, prompt-aligned videos that adapt to diverse lighting conditions, including dramatic spotlight effects.

"A dark wooden desk, ···, soft-edged white beams overlapping on a cyan backdrop"

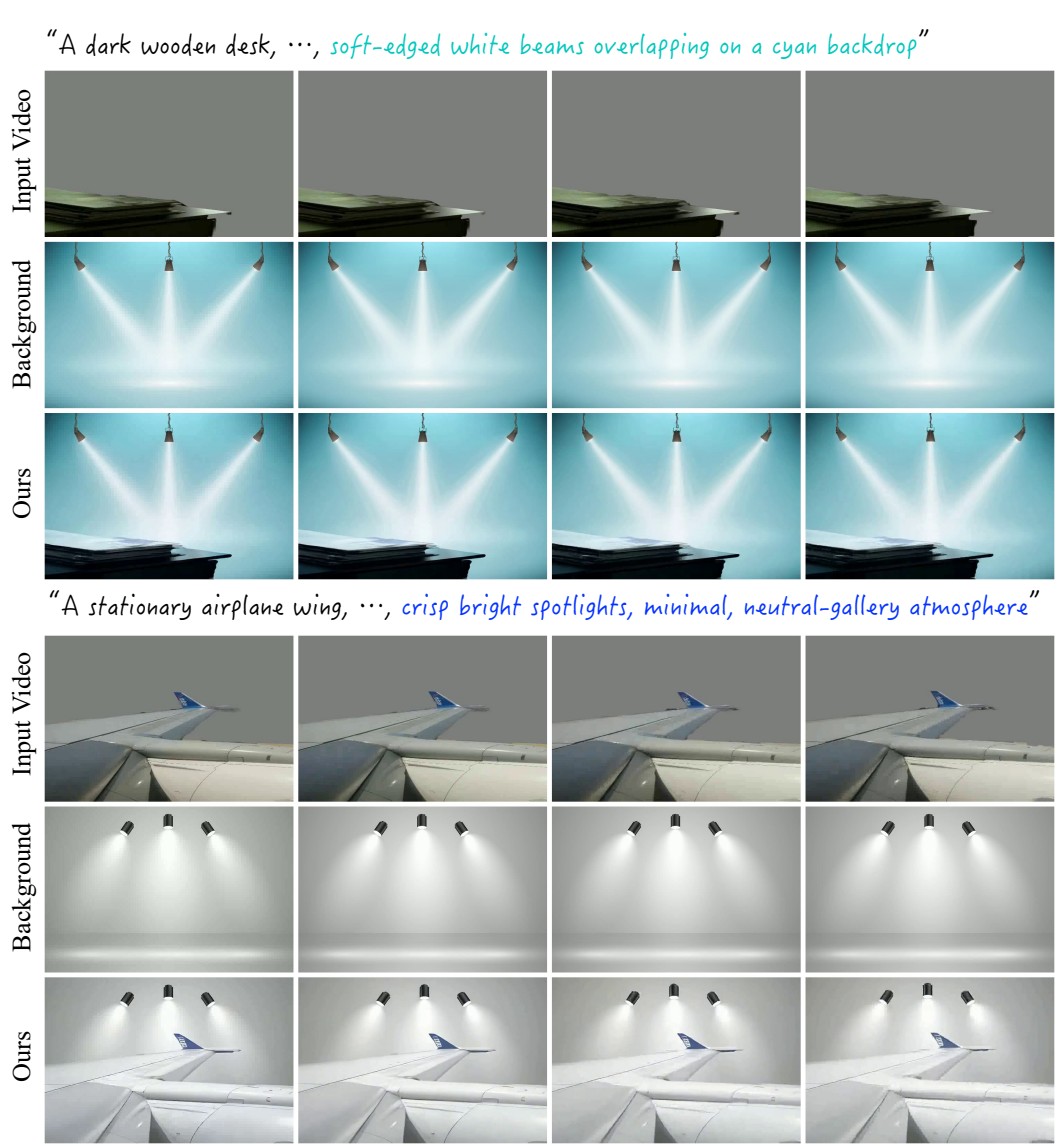

"A stationary airplane wing, ···, crisp bright spotlights, minimal, neutral-gallery atmosphere"

Figure 13: **Visual results of IllumiCraft.** Our method produces high-fidelity, prompt-aligned videos that adapt to diverse lighting conditions, including dramatic spotlight effects.

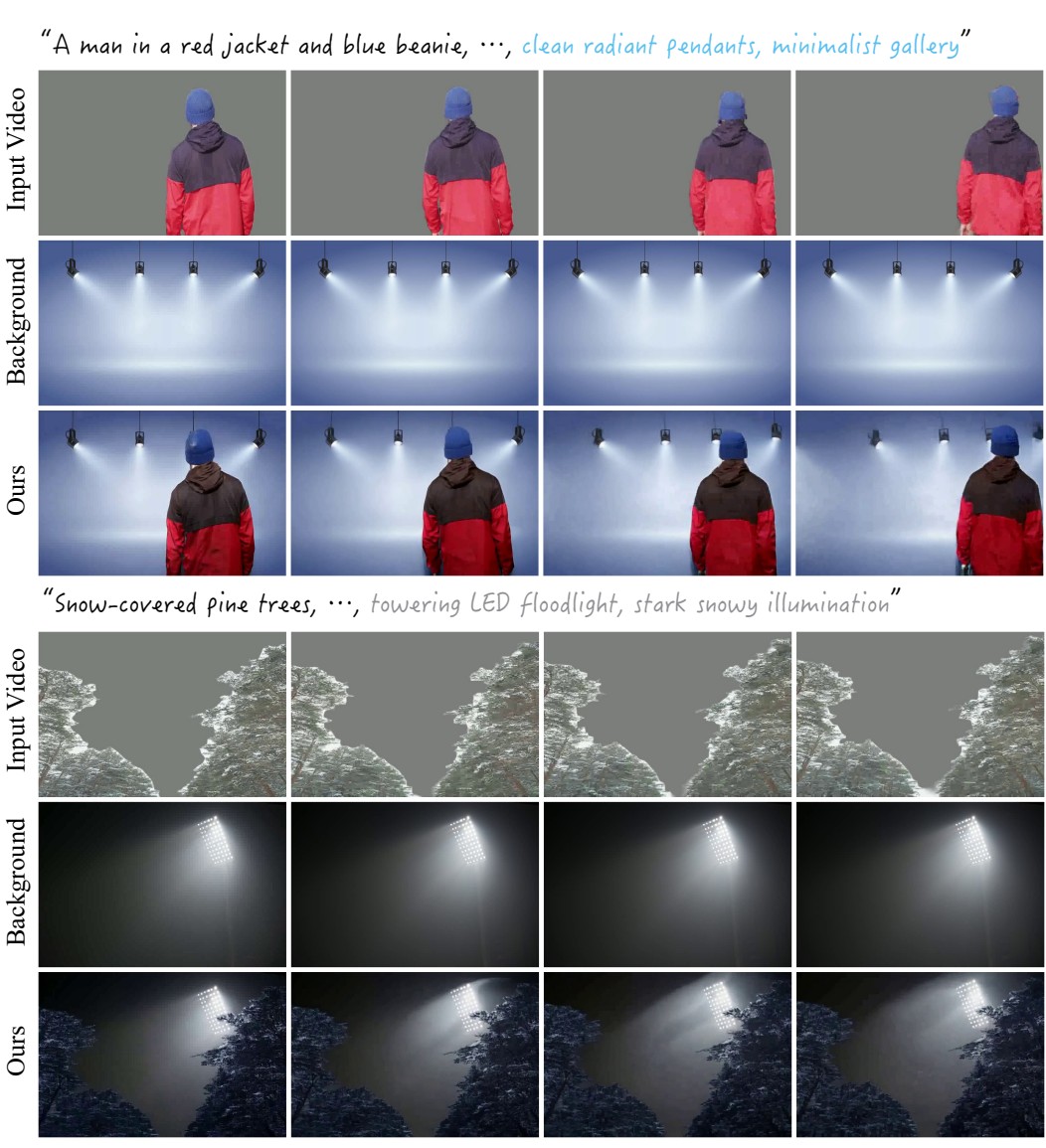

Figure 14: **Visual results of IllumiCraft.** Our method produces high-fidelity, prompt-aligned videos that adapt to diverse lighting conditions, including dramatic spotlight effects.

Table 11: Our collected lighting prompts (1-50) for relit videos.

| # | Lighting Prompt |
| --- | --- |
| 1 | red and blue neon light |
| 2 | sunset over sea |
| 3 | sunlight through the blinds |
| 4 | in the forest, magic golden lit |
| 5 | backlighting |
| 6 | sunset |
| 7 | sunshine, hard light |
| 8 | dappled light |
| 9 | magic lit, sci-fi RGB glowing, key lighting |
| 10 | neon light |
| 11 | magic golden lit |
| 12 | shadow from window, sunshine |
| 13 | sunlight through the blinds |
| 14 | neon light |
| 15 | cozy bedroom illumination |
| 16 | natural lighting |
| 17 | soft lighting |
| 18 | candle light |
| 19 | pink neon light |
| 20 | sunlit |
| 21 | warm sunshine |
| 22 | warm yellow and purple neon lights |
| 23 | neon, Wong Kar-wai, warm |
| 24 | cyberpunk style and light |
| 25 | yellow and purple neon lights |
| 26 | sunshine from window |
| 27 | moon light |
| 28 | soft sunshine |
| 29 | dark shadowy light |
| 30 | neo punk, city night |
| 31 | cyberpunk |
| 32 | golden hour light |
| 33 | blue hour lighting |
| 34 | tungsten light |
| 35 | fluorescent office lighting |
| 36 | street light at night |
| 37 | studio spotlight |
| 38 | rim light on subject |
| 39 | bokeh city light at night |
| 40 | TV screen glow in dark room |
| 41 | modern minimalistic LED glow |
| 42 | ambient underlit glow |
| 43 | soft dusk lighting |
| 44 | harsh industrial light |
| 45 | warm ambient room light |
| 46 | icy blue fluorescent glow |
| 47 | mystical twilight shimmer |
| 48 | low-key candlelight |
| 49 | rainy city neon |
| 50 | glowing backlight |

Table 12: Our collected lighting prompts (51-100) for relit videos.

| # | Lighting Prompt |
|---|---|
| 51 | strong lighting |
| 52 | rustic lantern light |
| 53 | overcast day glow |
| 54 | golden twilight shimmer |
| 55 | dull stormy light |
| 56 | subtle overhead illumination |
| 57 | steely warehouse lighting |
| 58 | vintage street lamp |
| 59 | cool-blue spotlights |
| 60 | glowing river reflections |
| 61 | sunburst window light |
| 62 | morning haze glow |
| 63 | afterglow silhouette |
| 64 | dawn light shadows |
| 65 | broken neon flicker |
| 66 | sleek futuristic luminescence |
| 67 | soft pastel glow |
| 68 | urban dusk illumination |
| 69 | underwater blue light |
| 70 | interior design spotlight |
| 71 | backlit street sign |
| 72 | glowing neon arches |
| 73 | morning sunbeam |
| 74 | rustling leaves with sun rays |
| 75 | subdued candlelit ambiance |
| 76 | serene twilight light |
| 77 | prismatic light effects |
| 78 | diffuse daylight |
| 79 | geometric LED array |
| 80 | rain-soaked neon reflections |
| 81 | subterranean light glow |
| 82 | bright white spotlights |
| 83 | dazzling sun flare |
| 84 | vibrant festival lights |
| 85 | enchanting aurora borealis |
| 86 | subtle office glow |
| 87 | twinkling fairy lights |
| 88 | chrome and neon reflections |
| 89 | fiery red spotlight |
| 90 | icy neon glow |
| 91 | sun-dappled forest light |
| 92 | electric dreamscape glow |
| 93 | irradiated room ambiance |
| 94 | scattered light beams |
| 95 | colored spectrum radiance |
| 96 | mystic foggy illumination |
| 97 | urban jungle glow |
| 98 | warm campfire light |
| 99 | bioluminescent glow |
| 100 | caustic rippling light |

