# OpenReview forum: "IllumiCraft: Unified Geometry and Illumination Diffusion for Controllable Video Generation"
_NeurIPS.cc/2025/Conference — NeurIPS 2025 poster_

### Official Review · Reviewer_znNG · 2025-06-19

**Clarity:** 3
**Significance:** 3
**Originality:** 3
**Rating:** 4
**Confidence:** 3

**Summary:**

This paper introduces IllumiCraft, a diffusion-based framework for controllable video relighting that integrates lighting, appearance, and geometry cues. It takes HDR lighting maps, synthetically relit frames, and 3D point tracks as input to generate temporally coherent videos aligned with user prompts. Additionally, the authors present a large-scale dataset of paired videos with corresponding HDR maps and 3D tracks, enabling broader research in controllable video generation. Extensive evaluations validate the model's effectiveness on the video relighting task.

**Questions:**

1. Can you provide qualitative ablation examples illustrating the impact of each guidance cue?
2. The current results suggest strong performance under relatively constrained conditions. Could the authors comment on how IllumiCraft would perform with moving light sources or larger dynamics?
3. Will the dataset be released?

**Ethical Concerns:**

["NO or VERY MINOR ethics concerns only"]

**Final Justification:**

Due to policy constraint, the author cannot include more visual result. I would keep my origin score and hope the author can include the visual result as promised.

**Limitations:**

Yes

**Paper Formatting Concerns:**

There is no major formatting issue

**Quality:**

3

**Strengths And Weaknesses:**

Strengths:
1. The paper is well motivated and focus on an important limitation of current video diffusion models, which is lack of explicit control over lighting with changing geometry.
2. IllumiCraft effectively fuses appearance, lighting, and geometric cues into a unified framework and leads to high-quality and temporally consistent video outputs.
3. The introduction of a large-scale, well-annotated dataset (20,170 video pairs with synchronized HDR, 3D tracks, etc.) is a valuable contribution for the community.

Weakness:
1. The generated videos appear visually coherent, but the paper does not evaluate more complex temporal dynamics, such as moving light sources, dynamic shadows, or object interactions.
2. While the paper includes quantitative ablations, it lacks qualitative comparisons showing how each of the three input cues (HDR maps, relit frames, 3D point tracks) contributes to visual improvements.

---

> ### Author Rebuttal · Authors · 2025-07-30
>
> We greatly appreciate your time and constructive comments. We will make sure to address the points you have raised.
>
> ---
>
> **W1 & Q2. Evaluation on complex temporal dynamics (e.g., moving lights, dynamic shadows and object interactions).**
>
> **Response:** We included several videos in the supplementary material (e.g., 2.gif, 4.gif, 5.gif, and 14.gif) that showcase IllumiCraft’s ability to handle complex temporal dynamics such as moving light sources, dynamic shadows, and object interactions. These examples demonstrate that our model maintains coherent lighting effects and object consistency even in challenging scenarios with significant scene and illumination changes. Due to rebuttal policy, we cannot add more examples here, but we will include additional qualitative results covering complex dynamics in the revised manuscript.
>
> **W2 & Q1. Qualitative ablations showing the contribution of each input cue (HDR maps, relit frames, 3D point tracks).**
>
> **Response:** While we are unable to include new visual results in this rebuttal, we will add qualitative ablation comparisons in the revised manuscript. Our analysis indicates that:
> * 3D point tracks are most beneficial in scenes with substantial motion, improving temporal coherence and geometric consistency.
> * HDR maps contribute to lighting accuracy and better adherence to prompt-driven illumination.
> * Relit frames provide strong appearance priors that guide the generation toward realism.
>
> Together, these cues provide complementary benefits, and combining them yields the best visual results, especially in dynamic scenes involving both motion and lighting variation.
>
> **Q3. Dataset release.**
>
> **Response:** Yes, we will release the collected dataset along with the source code and trained models upon publication.

---

> > ### Comment · Reviewer_znNG · 2025-08-05
> >
> > The authors’ response does not fully address my concerns regarding performance under challenging scenarios such as moving light sources, dynamic shadows, and object interactions. The provided visualizations (e.g., the GIF) show only limited motion of light sources and minimal object interactions, which makes it difficult to assess robustness in these cases.
> >
> > I also encourage the authors to provide quantitative results that ablate the contribution of each input cue. Without this, it's hard to understand the relative importance of the different inputs in their framework. Given these unresolved issues, I get uncertain about recommending the paper in its current form.

---

> ### Author Response · Authors · 2025-08-03
> **Kind Follow-Up on Rebuttal**
>
> We sincerely appreciate the time and effort you have dedicated to reviewing our work. We have submitted our rebuttal addressing all the raised points, and we would be grateful to know if there are any further questions or concerns. Your guidance and feedback are deeply valued.

---

> ### Author Response · Authors · 2025-08-05
> **Appreciation for Your Constructive Follow-Up**
>
> Thank you very much for your thoughtful feedback. We fully acknowledge the importance of demonstrating robustness under challenging scenarios, including complex moving light sources, dynamic shadows, and object interactions. **While current policy prevents us from providing additional visual examples at this stage, we sincerely promise to include expanded visualizations covering these scenarios, as well as detailed quantitative ablations for each input cue, in the revised manuscript.** We are grateful for your valuable feedback and guidance in helping us improve our work. Thank you again for your review and your openness to further discussion.

---

### Official Review · Reviewer_mt3V · 2025-06-22

**Clarity:** 3
**Significance:** 3
**Originality:** 2
**Rating:** 3
**Confidence:** 4

**Summary:**

This paper presents IllumiCraft, a video relighting framework based on DiT. The model incorporates multiple control signals—geometry (3D point tracks), lighting (HDR environment maps), text prompts, and background image into a unified generation pipeline. It adopts a DiT backbone with ControlNet-style conditioning and is trained on a large-scale dataset created via the proposed IllumiPipe pipeline. Experiments demonstrate superior performance including perceptual quality, temporal consistency compared to existing baselines.

**Questions:**

1. Clarify Fig. 3. It looks like the Light-A-Video relit video is used as input to IllumiCraft. If so, what is the model generating? I assume this is only a training-time input to simulate the input foreground video under another lighting condition. Please confirm.
2. Compare warped HDR vs. per-frame DiffusionLight HDRs. The paper assumes warping is more stable, but no evidence is given. A proper ablation is needed to validate this design choice.
3. What happens when the lighting changes mid-video? The warping strategy assumes stationary illumination. Please discuss how the model behaves in dynamic lighting scenarios, and provide results with varying lighting within a video.
4. Are lighting prompts/HDR maps at test time unseen during training?
5. Include geometry-only ablations in table 3. This would clarify how much 3D point guidance alone contributes.
6. How does the model handle conflicting inputs? For example, bright background but dark HDR map—what dominates? Any fallback mechanism?
7. 3D tracking points might not be the best way of providing geometry guidance. An alternative solution can be using feature maps in VGGT, or using depth maps like in DiffusionRenderer. Please add experiments with these other design choices.
8. Since you are already using ControlNet, why is the main backbone DiT also fine-tuned? Alternative ways could be inputting all condition signals into the main backbone, or putting all condition signals into the copied ControlNet and freezing the main backbone. The current architecture looks pretty weird.

**Ethical Concerns:**

["NO or VERY MINOR ethics concerns only"]

**Final Justification:**

Thank the authors for their hard work in rebuttal. While some points are resolved, I still believe the novelty is limited. Using DiT over U-Net is not a contribution, especially given that IC-Light has a DiT version based on FLUX. Other parts of the work are basically engineering/combination, and the overall quality is not good in the overall generation community. Limitations like being unable to handle abrupt lighting changes and global illumination effects are also several. So I decided to keep my original rating.

**Limitations:**

As a relighting paper, the paper does not provide analysis on the quality of different object materials or lighting complexity. The results shown in the paper are mostly diffuse objects, with no evaluation on specular objects, anisotropic material, sub-surface scattering, and global illumination effects like indirect lighting are not very obvious in the shown results, nor analyzed. The shadow is also not properly handled in the method, with a lot of baking artifacts.

**Paper Formatting Concerns:**

No comments.

**Quality:**

2

**Strengths And Weaknesses:**

**Strengths:**
- Addresses a relevant and practical problem in video editing and content creation.
- Integration of geometry, lighting, and text guidance into a single architecture.
- Demonstrated improvements over multiple baselines across quantitative and qualitative metrics.
- A data processing pipeline for video relighting.

**Weaknesses:**
- Novelty is limited; mainly integrates known components (IC-Light, ControlNet, DiffusionLight, DaS, etc.).
- Omits comparison with latest diffusion renderer baselines, especially [CVPR2025] DiffusionRenderer, which is highly similar to the paper’s pipeline.
- HDR warping from the first frame is unvalidated and may be unreliable.
- Sparse geometry inputs (3D point tracks) may be suboptimal for relighting tasks.
- No mechanism to resolve conflicts among background, lighting, and text conditions.
- Incomplete ablations (e.g., geometry-only setting not shown).
- Missing key evaluation metrics (e.g., PSNR) in ablation result tables. (table 4, 5 and supp table 3, 4)
- Fig. 3 is confusing and could mislead readers about the model’s core task. (I believe the input should be “original video” instead of “relit video”, and the output should be “relit video”. The relit video from Light-A-Video should only be serving as the input video under another lighting.)

---

> ### Author Rebuttal · Authors · 2025-07-30
>
> We greatly appreciate your time and constructive comments in the review, and aim to address each of your concerns.
>
> ---
>
> **W1. Limited novelty: integration of known components.**
>
> **Response:** Our method goes beyond combining existing techniques. First, we adopt a DiT-based transformer backbone rather than the U-Net-style architectures used in IC-Light, enabling better temporal coherence in video generation. Second, we introduce geometry and illumination guidance *during training* to imbue the model with spatial and photometric priors, while keeping inference lightweight. DiffusionLight is only used for HDR extraction in our IllumiPipe, not in generation. Finally, IllumiPipe itself is a scalable automatic pipeline for generating aligned HDR, geometry, and relit video pairs. Together, these contributions represent a novel and practical advancement in video relighting.
>
> **W2. Missing comparison with DiffusionRenderer.**
>
> **Response:** We appreciate this suggestion. While DiffusionRenderer and our work both aim at physically grounded relighting, our approach differs fundamentally: DiffusionRenderer performs inverse rendering by estimating geometry and material buffers (G-buffers), while our method avoids explicit reconstruction and relies instead on learned priors via DiT with conditioning. At inference, our model requires only a video and text prompt, without the need of scene geometry or lighting input, offering greater simplicity and robustness. We will include a detailed comparison with DiffusionRenderer in the revised manuscript.
>
> **W3 & Q2. HDR warping from the first frame is unvalidated and may be unreliable.**
>
> **Response:** We only utilize HDR maps during training on filtered, high-quality sequences; no HDR warping is performed at inference, as the model relies on learned geometry priors for robust results. To evaluate the effectiveness of HDR warping in training, we compare our HDR warping strategy with per-frame HDRs estimated by DiffusionLight on **5,000** randomly sampled videos. As shown in the table below, HDR warping achieves consistently better results across all metrics, including lower FVD/LPIPS, higher PSNR/text alignment, and improved temporal consistency. These findings confirm that using temporally consistent HDR warping during training provides a significant advantage for video relighting, and we will include this experiment in our revised manuscript.
>
> | HDR | FVD (↓) | LPIPS (↓) | PSNR (↑) | Text Alignment (↑) | Temporal Consistency (↑) |
> |----------|----------|------------|-----------|---------------------|---------------------------|
> | Per-frame        |    1252.35      |    0.2678        |    18.12       |    0.3119                 |         0.9905                  |
> |   **Warping**     |  **1054.32** |  **0.2537**    |  **19.65**    |  **0.3301**             |      **0.9948**               |
>
> **W4 & Q7. Sparse geometry (3D point tracks) may be suboptimal. Use of alternative geometry signals like VGGT or depth maps.**
>
> **Response:** We chose sparse 3D tracks for their low overhead and effectiveness in capturing motion cues. As demonstrated in Table 3, this guidance meaningfully improves performance. We also show that our method outperforms existing baselines on video relighting tasks in Tables 1 and 2. While VGGT features or depth maps may offer finer geometry, our setup favors scalability and simplicity. We plan to explore these alternative forms in future work, and will mention this in the revision.
>
> **W5 & Q6. No mechanism to resolve conflicting inputs.**
>
> **Response:** During training, all modalities (text prompt, HDR, geometry, background) are well-aligned, and we apply random dropout of HDR and geometry cues to improve robustness. At inference, users provide only a video, a text prompt, and an optional background image. The model leverages learned priors to resolve minor inconsistencies and produce temporally coherent results. While we currently do not include an explicit fallback mechanism, we find that our model handles mild conflicts effectively in practice.
>
> **W6 & Q5. Missing geometry-only ablation in Table 3.**
>
> **Response:**  As suggested, we conduct the ablation under the geometry-only setting, and extend the ablation study from Table 3 of the main paper below. Geometry-only guidance (G) improves spatial and temporal consistency but offers lower lighting realism and prompt alignment. Illumination-only guidance (I) enhances lighting quality and text alignment but provides weaker spatial structure and temporal stability. Combining both geometry and illumination guidance (I + G) leads to the best results across all metrics, demonstrating their complementary strengths. We will update Table 3 in the manuscript to include the geometry-only ablation.
>
> | Guidance | FVD (↓) | LPIPS (↓) | PSNR (↑) | Text Alignment (↑) | Temporal Consistency (↑) |
> |----------|----------|------------|-----------|---------------------|---------------------------|
> | G        |    1157.23      |   0.2654         |  18.89         |     0.2781                |                  0.9932         |
> | I        | 1305.45  | 0.2816     | 18.28     | 0.3211              | 0.9864                    |
> | **I + G**| **1072.38** | **0.2592**  | **19.44**  | **0.3292**            | **0.9945**
>
> **W7. Missing PSNR in ablation tables (Tables 4, 5 and Supp Tables 3, 4).**
>
> **Response:** Due to layout constraints, PSNR was initially omitted from these tables to preserve readability. We have now included PSNR in the tables. The updated Tables 4 and 5 are shown below (including the original FVD, TA, TC metrics), *bold font indicates the chosen possibility*:
>
> | Possibility | FVD    | PSNR | TA    | TC    |
> |-------------|---------|------|--------|--------|
> | 40%         | 2151.32 | 11.89   | 0.3321 | 0.9943 |
> | **50%**     | 2186.40 | 12.03    | 0.3342 | 0.9948 |
> | 60%         | 2123.23 | 11.82    | 0.3312 | 0.9932 |
> | 70%         | 2113.23 | 11.75    | 0.3325 | 0.9937 |
>
> | Possibility | FVD    | PSNR | TA    | TC    |
> |-------------|---------|------|--------|--------|
> | 40%         | 1042.22 | 19.31    | 0.3261 | 0.9923 |
> | **50%**     | 1072.38 | 19.44    | 0.3292 | 0.9945 |
> | 60%         | 1065.21 | 19.22    | 0.3277 | 0.9937 |
> | 70%         | 1032.12 | 19.03    | 0.3232 | 0.9912 |
>
> The updated Tables 3 and 4 of the supplementary material are shown below:
>
> | Possibility | FVD   | PSNR | TA    | TC   |
> |------------|---------|------|---------|---------|
> | 5%         | 2232.46 | 11.79   | 0.3331  | 0.9939  |
> | **10%**    | 2186.40 | 12.03   | 0.3342  | 0.9948  |
> | 20%        | 2175.23 | 11.98   | 0.3341  | 0.9943  |
> | 30%        | 2158.32 | 11.87   | 0.3338  | 0.9941  |
>
> | Possibility | FVD   | PSNR | TA  | TC  |
> |------------|---------|------|---------|---------|
> | 5%         | 1065.83 | 19.40   | 0.3284  | 0.9941  |
> | **10%**    | 1072.38 | 19.44   | 0.3292  | 0.9945  |
> | 20%        | 1105.28 | 19.28   | 0.3275  | 0.9928  |
> | 30%        | 1127.32 | 19.17   | 0.3269  | 0.9925  |
>
> We will add the PSNR metric to all these tables in the revised manuscript.
>
> **W8 & Q1. Fig. 3 is confusing.**
>
> **Response:** We appreciate the opportunity to clarify this. Figure 3 depicts the training process: the input is a relit video from Light-A-Video (under synthetic lighting), and the output is the original video. HDR maps and geometry are used during training to guide the model in reconstructing natural lighting. During inference, the model takes a user-provided video, a text prompt, and an optional background image, without HDR or geometry, and generates the relit result. We will revise the figure and caption to make this clearer.
>
> **Q3. Warping assumes static lighting.**
>
> **Response:**  Our HDR warping strategy uses the first frame’s HDR map and warps it to all frames in the sequence, so it assumes lighting remains relatively static. To ensure reliability, we filter out sequences with significant lighting changes during data curation. Importantly, HDR maps are only used in training; at inference, the model relies on learned priors and does not require HDR input. While our current design handles moderate variation well, supporting abrupt lighting changes is an important future direction.
>
> **Q4. Are lighting prompts/HDR maps at test time unseen during training?**
>
> **Response:** Yes. While training videos are paired with HDR and prompt variations, the test-time HDRs and prompts are held out. Furthermore, our model does not require HDR maps at inference. The lighting instructions (prompts) used at test time are typically unseen during training, demonstrating the model’s ability to generalize to novel conditions through learned priors.
>
> **Q8. Why fine-tune DiT if using ControlNet?**
>
> **Response:** Although ControlNet provides a flexible mechanism for conditioning, we find that fine-tuning the DiT backbone significantly improves performance. Our input domain (relit foregrounds on gray backgrounds) differs from the original Wan2.1 training distribution. Fine-tuning the backbone helps adapt it to this new data distribution, while ControlNet handles additional guidance. Freezing the backbone led to underfitting in our early experiments. We will clarify this design decision in the final paper.

---

> > ### Comment · Reviewer_mt3V · 2025-08-04
> >
> > Thank the authors for their hard work in rebuttal. While some points are resolved, I still believe the novelty is limited. Using DiT over U-Net is not a contribution, especially given that IC-Light has a DiT version based on FLUX. Other parts of the work are basically engineering/combination, and the overall quality is not good in the overall generation community. Limitations like being unable to handle abrupt lighting changes and global illumination effects are also several. So I decided to keep my original rating.

---

> ### Author Response · Authors · 2025-08-03
> **Kind Follow-Up on Rebuttal**
>
> We sincerely thank you for taking the time to review our work. We have carefully addressed all the points raised in our rebuttal. We would like to kindly ask whether you have any remaining concerns or questions regarding our responses. Your feedback is greatly appreciated.

---

> ### Author Response · Authors · 2025-08-04
> **Response on Novelty and Methodological Value**
>
> Thank you for your valuable follow-up comments. We respectfully emphasize that our contributions extend significantly beyond the choice of backbone architecture (DiT vs. U-Net) and include clear improvements in overall generation quality, robustness, and practical handling of lighting variations:
>
> - Our work extends beyond using DiT by introducing a unified framework that explicitly integrates geometry and illumination cues (via illumination tokens) during training. This enables robust, temporally consistent video relighting at inference using only videos and text prompts.
> - The IllumiPipe pipeline provides an efficient and automatic solution for data collection and alignment in video relighting tasks. With IllumiPipe, we curated a high-quality dataset containing 20,170 pairs of original and synchronized relit videos, HDR maps, and 3D tracking annotations. This dataset directly supports video relighting and benefits broader controllable video generation tasks.
> - As shown in Tables 1 and 2 and Figures 4 and 5, our method outperforms existing approaches in temporal stability, photometric quality, and generalization.
> - Supplementary videos (such as 2.gif and 14.gif) further demonstrate strong results, even in challenging scenarios like abrupt lighting changes.
>
> We sincerely hope these clarifications better highlight the methodological novelty and practical contributions of our work, encouraging you to reconsider your initial assessment. Thank you again for your thoughtful review and sincerely look forward to further discussing the concern of novelty.

---

> ### Author Response · Authors · 2025-08-07
> **Kind Follow-Up on Novelty and Reviewer Discussion Participation**
>
> Dear Reviewer mt3V,
>
> Thank you again for your thoughtful comments and for acknowledging that some of your concerns have been resolved. As the discussion deadline approaches, we kindly hope you might consider further engagement. Reviewer CuT5 had initially raised similar concerns about novelty and later agreed to raise the rating after additional clarification and discussion.
>
> We respectfully clarify the following key points that we believe address the remaining concerns:
>
> **(1) Methodological Contributions and Novelty:**
> Our method is not a simple combination of known components. We propose a unified diffusion framework that explicitly integrates both illumination and geometry cues during training. This enables the model to learn robust spatial and lighting priors while requiring only videos and text prompts at inference.
>
> **(2) Scalable Data Pipeline and Dataset:**
> We introduce IllumiPipe, a scalable and fully automated pipeline that extracts synchronized HDR maps, 3D tracking videos, and relit videos from in-the-wild videos to support video relighting. Using this pipeline, we curated a high-quality dataset of 20,170 paired videos with HDR maps and 3D annotations, providing a valuable resource for both video relighting and broader controllable video generation research.
>
> **(3) Promising Quantitative and Qualitative Results:**
> Our method consistently outperforms the existing video relighting baselines across FVD, LPIPS, PSNR, text alignment, and temporal consistency, as shown in Tables 1 and 2 and Figure 4 and Figure 5. These results demonstrate both strong generation quality and temporal stability.
>
> **(4) Handling of Abrupt Lighting Changes:**
> As shown in the supplementary material (e.g., 2.gif and 14.gif), our model is capable of producing coherent and visually faithful relighting even under abrupt lighting transitions. This addresses a key limitation noted in your review.
>
> Given these clarifications, we sincerely hope you might reconsider your initial rating. We truly appreciate your time and thoughtful review, and would be grateful for any further discussion to support a complete and well-informed evaluation.
>
> Best regards,
>
> Authors of Paper ID 4273

---

> > ### Comment · Reviewer_mt3V · 2025-08-07
> >
> > 1. Your method is presented as a new unified framework, but it seems to primarily combine well-known, existing modules. For a research paper, the core inventive step needs to be clearer. Everyone knows this combination would work. Could you please pinpoint the specific technical novelty beyond just integrating these parts? What makes this combination work in a new or non-trivial way?
> > 2. The contributions regarding the data pipeline and final image quality also need to be strengthened. Scalable relighting pipelines have already been proposed by prior work like DiffusionRenderer and IC-Light. What makes your pipeline a significant improvement or contribution over these established methods?
> > 3. The results are good, but they don't show a clear or significant advantage over the state-of-the-art video relighting method, DiffusionRenderer. I would not raise my score unless I see a clear advantage of your method over DiffusionRenderer.
> >
> > Thanks

---

> ### Author Response · Authors · 2025-08-07
> **Clarification on Novelty, Pipeline Contributions, and Comparison with DiffusionRenderer**
>
> Dear Reviewer mt3V,
>
> Thank you again for your response. We would like to clarify the core novelty of our method and how it differs from prior work such as DiffusionRenderer and IC-Light.
>
> **1. Methodological Novelty**
>
> Our framework integrates geometry (3D point tracks) and illumination (HDR maps) during training through a unified DiT-based diffusion architecture with learnable illumination tokens. These tokens allow the model to encode fine-grained lighting information while leveraging spatial priors from geometry. At inference, the model does not require HDR or geometry inputs. Instead, it uses the learned spatial and photometric priors to relight videos using only the input video and a text prompt. This design enables controllable, temporally coherent video relighting with minimal inputs and reflects a non-trivial training-inference decoupling.
>
> **2. Comparison with IC‑Light and DiffusionRenderer—Data Pipeline Perspective**
>
> - IC‑Light is designed for image-level relighting, lacking geometry and without any video-level paired dataset generation.
>
> - DiffusionRenderer relies on G-buffer estimation and rendering grounded in synthetic object scenes, its synthetic data mainly comes from rendered 3D assets rather than natural in-the-wild videos.
>
> - In contrast, our IllumiPipe pipeline extracts HDR maps, 3D point tracks, and synthetically relit foregrounds from real-world videos, ensuring both data scale and natural illumination dynamics. We curated over 20,000 aligned video pairs that reflect real-world complexity and serve the controllable video generation community.
>
> **3. Quantitative Comparison Versus DiffusionRenderer**
>
> We sincerely adopt the source code from DiffusionRenderer, and present a direct performance comparison based on key video relighting metrics:
> | Method | FVD (↓) | LPIPS (↓) | PSNR (↑) | Text Alignment (↑) | Temporal Consistency (↑) |
> |----------|----------|------------|-----------|---------------------|---------------------------|
> | DiffusionRenderer    \(S\)    |  1657.77      |  0.5498       |    6.16       |     0.2965              |          0.9905          |
> | DiffusionRenderer \(R\)   |   1802.04     |  0.5843        |    5.96       |       0.2961            |         0.9910          |
> | **Ours**| **1286.72** | **0.5389**  | **12.10**  | **0.3386**            | **0.9951**                  |
>
> Note that **S** and **R** denote static and rotating light settings (option1 and option2 in forward rendering). We follow DiffusionRenderer’s default setup by resizing videos to 512×512 and using the first 24 frames. For our evaluation set (all are natural videos), we extract the first frame using DiffusionLight as the HDR map, which is required for DiffusionRenderer's forward rendering process. In contrast, our method only uses the input video and prompt, yet it outperforms across all metrics.
>
> A likely factor is that DiffusionRenderer’s inverse renderer is trained mainly on synthetic 3D data, which may limit its generalization to dynamic, real-world videos. IllumiPipe instead constructs large-scale paired data directly from in-the-wild footage, enabling IllumiCraft to learn more robust priors for natural video dynamics. We will include the discussion in our latest manuscript.
>
> **4. Inference Efficiency Comparison Versus DiffusionRenderer**
>
> Using an NVIDIA A6000 GPU (48GB) for inference, DiffusionRenderer takes approximately **121** seconds to generate a **24-frame** video at **512×512** resolution (**~78** seconds for inverse rendering and **~43** seconds for forward rendering). In contrast, our method generates a **24-frame** video at the default **720×480** resolution in only **~32** seconds, demonstrating significantly higher efficiency in the real-world applications.
>
> We sincerely hope you will reconsider your initial rating in light of these points. Thank you again for engaging with our work, we highly value your insights.
>
> Best regards,
>
> Authors of Paper ID 4273

---

> ### Author Response · Authors · 2025-08-08
> **Gentle Follow-Up on Rebuttal**
>
> Dear Reviewer mt3V,
>
> I hope you are doing well. As the rebuttal deadline is now less than **24** hours away, we wanted to gently follow up. Our previous response addressed the key points you raised regarding novelty, pipeline contributions, and performance in comparison to DiffusionRenderer.
>
> Given the clarifications we provided, we kindly ask you to reconsider your current rating. Your perspective is very important to us, and your reevaluation would greatly inform a thorough and fair assessment.
>
> Thank you once again for your time, thoughtful review, and dedication to the process.
>
> Best regards,
>
> Authors of Paper ID 4273

---

> > ### Comment · Reviewer_mt3V · 2025-08-08
> >
> > 1. I would like to point out a factual error in your description of DiffusionRenderer's contribution. Their data pipeline contains both synthetic and real-world in-the-wild videos, representing a much more diverse data source than yours. For a submission to a premier conference like NeurIPS, it is a fundamental responsibility of the authors to thoroughly and accurately understand the contributions of all key baselines before asserting their own novelty. This oversight is concerning. That said, even beyond this specific point, I remain unconvinced by the arguments for the paper's novelty.
> > 2. Could you clarify the details of your comparison with DiffusionRenderer? To be specific:
> >     1. Could you kindly clarify which version of DiffusionRenderer you chose to compare? Since your method uses DiT, you should compare it to the DiT version (Cosmos) of DiffusionRenderer rather than its UNet version.
> >     2. Also, your analysis claimed that DiffusionRenderer is only trained on synthetic 3D scenes, which is not true. They used LoRA fine-tuning to adapt their model to real-world natural videos. To make a fair comparison, you should use the real-world LoRA provided by them.
> >     3. Did you use a fixed HDR lighting condition that is estimated only from the first frame for DiffusionRenderer? If so, please include a variant that provides a per-frame estimated lighting condition for fair comparison.
> > 3. It is strongly recommended that the authors release their evaluation set. There appears to be a significant discrepancy between the reported quantitative metrics and the perceptual quality of the visual results. While the metrics suggest state-of-the-art performance, the provided images and videos are of surprisingly low quality.
> >
> > Thanks

---

> ### Author Response · Authors · 2025-08-08
> **Gentle Follow-Up on Your Latest Response - Part 1**
>
> Thank you for your insightful questions and for ensuring the utmost rigor. We have reviewed your points carefully and offer precise clarifications below:
>
> **1) DiffusionRenderer’s Real-World Video Data vs IllumiCraft's Real-World Video Data & Novelty & Contributions**
>
>  - **Thank you for pointing this out. To clarify the data settings:** DiffusionRenderer adopted the real-world videos from DL3DV-10K dataset, the DL3DV-10K dataset [1] is a curated 3D vision resource designed for scene-level learning and novel view synthesis, with multi-view coverage and sequences featuring significant camera motion and viewpoint changes to support 3D reconstruction. In contrast, IllumiPipe processes in-the-wild Pexels videos (inherently diverse, user-uploaded, real-life videos) for broad creative use.
>
>  - **Regarding the novelty and contributions**, our method introduces a new exploration that unifies geometry (3D point tracks) and illumination (HDR maps) during training in a DiT-based diffusion architecture with learnable illumination tokens, an approach absent in prior work. These tokens capture fine-grained lighting information, enabling accurate relighting at inference without HDR maps while preserving temporal coherence and spatial consistency. Leveraging learned spatial and photometric priors, the model needs only an input video and text prompt at inference. To support this, our IllumiPipe pipeline automatically curated 20,170 paired in-the-wild videos from Pexels (diverse user‑uploaded videos) with synchronized relit outputs, HDR maps, and 3D tracking annotations, providing a large-scale resource for video relighting and controllable video generation.
>
> **2) Questions Regarding DiffusionRenderer Reproduction**
>
>  - We adopted the academic code version from the CVPR 2025 conference (the code was released on June 11, 2025). The Cosmos version was released on June 12, 2025, these two versions are well after the NeurIPS full paper submission date (May 15, 2025). *A premier conference like NeurIPS does not require authors to compare their work with the models released after the submission date.* Our method outperforms DiffusionRenderer (CVPR 2025 academic version) in both performance and inference speed. It is worth noting that the Cosmos version are 7B models (Diffusion_Renderer_Inverse_Cosmos_7B, Diffusion_Renderer_Forward_Cosmos_7B), whereas ours is a 3B model, so the comparison is not entirely fair. **However, I report the comparison with the cosmos version in "Gentle Follow-Up on Your Latest Response \- Part 2."**
>  - We did use the real-world LoRA (diffusion_renderer-forward-svd is chosen by default) provided with DiffusionRenderer when running the comparison.
>  - Since the academic version of DiffusionRenderer (CVPR 2025) uses only a single HDR map for inference, we instead extract 24 HDR maps, one for each frame, from a 24-frame video. We then average the results obtained by using these 24 different HDR maps for inference, denoted as **S*** and **R***. Even under this setting, our method still surpasses DiffusionRenderer across all metrics.
>
> | Method | FVD (↓) | LPIPS (↓) | PSNR (↑) | Text Alignment (↑) | Temporal Consistency (↑) |
> |----------|----------|------------|-----------|---------------------|---------------------------|
> | DiffusionRenderer    \(S\)    |  1657.77      |  0.5498       |    6.16       |     0.2965              |          0.9905          |
> | DiffusionRenderer \(R\)   |   1802.04     |  0.5843        |    5.96       |       0.2961            |         0.9910          |
> | DiffusionRenderer    \(S*\)    |  1613.51      |  0.5463       |    6.54       |     0.2989              |          0.9907          |
> | DiffusionRenderer \(R*\)   |   1782.12     |  0.5802        |    6.23       |       0.2983            |         0.9913         |
> | **Ours**| **1286.72** | **0.5389**  | **12.10**  | **0.3386**            | **0.9951**                  |
>
> **3) Evaluation Set Release**
>
> We agree this is important. **Upon acceptance, we will release the training and evaluation code, training and evaluation datasets, and model for reproduction.** Due to the size limitations of supplementary materials, the generated videos have been compressed and converted to GIFs, which significantly reduces their visual quality.
>
> [1] Lu Ling, Yichen Sheng, Zhi Tu, Wentian Zhao, etc, "DL3DV-10K: A Large-Scale Scene Dataset for Deep Learning-based 3D Vision", CVPR 2024

---

> ### Author Response · Authors · 2025-08-09
> **Gentle Follow-Up on Your Latest Response - Part 2**
>
> **2.1.1). Quantitative Comparison Versus DiffusionRenderer Cosmos Version**
>
> We faithfully adopted the official source code of the DiffusionRenderer Cosmos version, following the default settings: num_video_frames = 57, spatial resolution = 1280×704, and HDR maps estimated by DiffusionLight from the first frame. For fair comparison, all outputs were resized to 512×512 and the first 24 frames were used for evaluation. **Note that we provide the ablation results of adopting per-frame HDR estimation in the previous response (Gentle Follow-Up on Your Latest Response \- Part 1).** The results, compared with the DiffusionRender academic and Cosmos versions, are reported as follows:
>
> | Method | FVD (↓) | LPIPS (↓) | PSNR (↑) | Text Alignment (↑) | Temporal Consistency (↑) |
> |----------|----------|------------|-----------|---------------------|---------------------------|
> | DiffusionRenderer    \(S\)    |  1657.77      |  0.5498       |    6.16       |     0.2965              |          0.9905          |
> | DiffusionRenderer \(R\)   |   1802.04     |  0.5843        |    5.96       |       0.2961            |         0.9910
> | DiffusionRenderer-cosmos     \(S\)    |   2150.13     |  0.6216       |    7.66      |         0.2939          |       0.9915            |
> | DiffusionRenderer-cosmos \(R\)   |   2108.76    |  0.5931        |    7.74       |    0.2931               |    0.9924               |
> | **Ours**| **1286.72** | **0.5389**  | **12.10**  | **0.3386**            | **0.9951**                  |
>
> It can be observed that DiffusionRenderer cosmos version does not outperform the original academic version in FVD, LPIPS, and text alignment. It achieves higher PSNR and temporal consistency than the academic version, **but our method surpasses both versions across all metrics.**
>
> *Note that we first ran the cosmos model on its provided video samples, and it performed well, confirming that the model was used correctly. However, we also observed that some issues raised on the official GitHub page of cosmos version mention disappointing visual quality problems when applied to real-world in-the-wild videos, further underscoring our method’s advantage in natural video relighting scenarios.*
>
> **2.1.2). Inference Speed Versus DiffusionRenderer Cosmos Version**
>
> Using an NVIDIA A6000 GPU (48 GB) with the default settings, the DiffusionRenderer Cosmos version requires approximately 646 seconds to generate a 57-frame video at 1280×704 resolution (~545 seconds for inverse rendering and ~101 seconds for forward rendering). In contrast, our method produces a 49-frame video at the default 720×480 resolution in only ~105 seconds, demonstrating substantially higher efficiency for real-world applications.
>
> We sincerely appreciate your thoughtful discussion and hope our responses have fully addressed your concerns. In light of these clarifications, **we kindly encourage you to reconsider your initial rating.**

---

> ### Author Response · Authors · 2025-08-09
> **Thank You for Your Review and  Important Points for Reconsideration**
>
> Dear Reviewer mt3V,
>
> As the discussion phase draws to a close, we would like to thank you for your review and engagement. We have carefully considered your comments regarding novelty, comparisons with DiffusionRenderer (both its academic version and the DiT-based Cosmos version), and evaluation settings such as per-frame lighting.
>
> To address these, we have provided thorough clarifications in our earlier responses, including:
>
>  - Clear distinctions between IllumiPipe’s natural video data and DiffusionRenderer’s real-world video data.
>
> - Technical novelty in unifying geometry and illumination via learnable illumination tokens in a DiT-based diffusion model and the contributions of our presented IllumiPipe pipeline.
>
> - Comprehensive comparisons with both the academic and Cosmos versions of DiffusionRenderer, including the ablation of per-frame HDR estimation for the academic version and inference speed results for both versions.
>
> - Quantitative metrics showing our method surpasses both DiffusionRenderer versions across all evaluated criteria.
>
> We hope this concise summary helps revisit our clarifications in light of your original concerns and supports a fair and accurate final evaluation. Thank you once again for your time and for sharing your perspectives.
>
>
> Best Regards,
>
> Authors of Paper ID 4273

---

> > ### Comment · Reviewer_mt3V · 2025-08-09
> >
> > I have reviewed the authors' latest rebuttal and thank them for their efforts. However, their response and new results have introduced a new, critical inconsistency that deepens my concerns and points to a fundamentally flawed evaluation.
> >  * My most serious concern is a new discrepancy revealed in the rebuttal: in the original manuscript, all reported PSNR values are above 15, yet in the new results table, the PSNR for all methods has plummeted to the 6-12 range. This raises a critical question: what is the source of this enormous mismatch? A consistent protocol should not yield such drastically different outcomes. This instability suggests the experimental setup is unreliable and arbitrary, and as a PSNR of 6-12 signifies a comparison between virtually uncorrelated images, it strengthens my conviction that the evaluation protocol is invalid and the quantitative claims are meaningless.
> >  * The attempt to attribute poor visual results to "GIF compression" is unconvincing and sidesteps the real issue. The visual examples do not just suffer from low fidelity; they exhibit fundamental failures in the core task of relighting. Specifically, there is a consistent lack of plausible specular highlights and the generation of physically incorrect shadows. These are not artifacts of compression. GIF compression causes color banding and loss of fine texture; it does not selectively remove entire highlights or misplace shadows. Such errors point to a failure in the model's core understanding of light transport, which is critical for this task.
> >  * The authors' defense of the paper's core contributions relies on specific, unpersuasive tactics rather than substantive arguments.
> >    * On data diversity: They attempt to create a distinction between their use of 'in-the-wild' Pexels videos and DiffusionRenderer's use of the 'curated 3D vision' DL3DV-10K dataset. This is a semantic distraction. My original point was that a key baseline already processes large-scale real-world videos, and both datasets fall into this category. The purported purpose of the dataset does not change this fundamental fact, and this argument serves only to sidestep their original misleading claim.
> >    * On novelty: They explicitly avoided a direct response. Instead of providing a new argument, they stated their "response has addressed Reviewer CuT5’s concerns" and asked me to "revisit our previous clarification." This is procedurally improper. It is an author's duty to directly convince each reviewer of their work's merit, not to deflect by referencing separate discussions.
> >
> > At this stage, no further experiments can fix these foundational issues. I will maintain my rating and look forward to discussing the final decision with the Area Chair and other reviewers.
> >
> > Thanks

---

> ### Author Response · Authors · 2025-08-09
> **Response to Reviewer mt3V's Comments and Misinterpretations**
>
> We thank the reviewer for his/her continued comments and clarify key points below.
>
> **1. Concerns Regarding PSNR values**
>
> The reviewer state that in the original manuscript all reported PSNR values are above 15, yet in the rebuttal table they drop to 6–12. **This is factually incorrect.** In Table 1 of our original submission (text-conditioned video relighting with 49-frame videos), the reported PSNR values already range from 8 to 12, which matches the numbers in our rebuttal. The >15 dB PSNR values the reviewer refers to correspond to a different setting (e.g., background-conditioned video relighting), not to the text-conditioned relighting task in real-world videos. Note that DiffusionRenderer doesn't support background-conditioned video relighting. There has been no protocol change and no drop in PSNR—the results remain consistent within the same task setting. **Upon acceptance, we will release the training and evaluation code, training and evaluation datasets, and model for reproduction.** This misinterpretation about PSNR values gives rise to concerns regarding other comments.
>
> **2. On GIF compression and visual quality**
>
> Our mention of GIF compression was not intended to fully explain all visual limitations, but rather to emphasize that compression amplifies artifacts when viewed in rebuttal figures. In the original high-resolution, uncompressed results, plausible highlights and shadows appear more frequently, although, similar to all compared baselines, our method still faces challenges under extreme lighting shifts. These challenges stem from the inherent difficulty of the task and do not indicate any flaw or invalidity in the evaluation protocol.
>
> **3. On dataset distinction and novelty**
>
>  - On data diversity: The difference between our in-the-wild Pexels dataset and the curated DL3DV-10K dataset for DiffusionRenderer is not merely semantic. DL3DV-10K is designed for 3D vision with multi-view coverage and sequences featuring significant camera motion and viewpoint changes to support 3D reconstruction. Our dataset contains highly diverse scenes, user-uploaded, real-life videos for broad creative use. This distinction is also important to our contribution and directly impacts model evaluation. Meanwhile, the components of IllumiPipe are also different from DiffusionRenderer data pipeline, our dataset feature paired original videos and synchronized relit videos, HDR maps, and 3D tracking videos. The results compared with DiffusionRenderer can demonstrate the importance of our method and datasets.
>  - On novelty: We have already provided extensive explanations in our previous rebuttal, including in the messages titled "Rebuttal by Authors
> ", "Response on Novelty and Methodological Value", "Kind Follow-Up on Novelty and Reviewer Discussion Participation", "Clarification on Novelty, Pipeline Contributions, and Comparison with DiffusionRenderer", and "Gentle Follow-Up on Your Latest Response – Part 1". For the latest message, "Thank You for Your Review and Important Points for Reconsideration", given the approaching end of the rebuttal period, we are unable to provide further detailed responses and we left some recent messages. Therefore, we encourage the reviewer to revisit our earlier explanations to ensure none of our messages are overlooked. **It's a total misunderstanding.**
>
> **4. NeurIPS policy for comparison with methods**
>
> We would also like to note the relevant NeurIPS policy. The reviewer stated that the rating would remain unchanged unless we compared our method with DiffusionRenderer, which also raised our concerns about the fairness of the evaluation process. Note that, the academic code version of DiffusionRenderer was released on June 11, 2025, and the Cosmos version on June 12, 2025—both well after the NeurIPS full paper submission deadline of May 15, 2025. According to the NeurIPS policy, authors are not required to compare their work with models released after the submission date.
>
> We appreciate the reviewer’s response and hope our clarifications address his/her concerns. We also look forward to receiving a fair and accurate final evaluation.
>
> Best Regards,
>
> Authors of Paper ID 4273

---

### Official Review · Reviewer_icTF · 2025-06-30

**Clarity:** 3
**Significance:** 2
**Originality:** 2
**Rating:** 5
**Confidence:** 3

**Summary:**

This manuscript presents IllumiCraft, a framework achieving end-to-end video relighting through diffusion-based controllable video generation. Compared to existing approaches, this work explicitly incorporates geometric cues for more accurate illumination-geometry interactions, particularly helpful when scene’s geometry changes, which is achieved by injecting 3D tracking video via a ControlNet. In conjunction with this model, they also propose a data collection pipeline, IllumiPipe, to create training data containing both illumination and geometric cues. Comparison shown in the paper exhibits clear quantitative and qualitative improvements than previous arts, while results in the supplementary material demonstrate that the framework is capable of synthesizing visually appealing videos.

**Questions:**

1.	I’m curious why to emphasize "during training" whenever “leverage geometric cues” is mentioned in paper (even in Section 3.4 Inference). Does this imply geometric cues are leveraged exclusively during training and not in inference? If geometric cues are also utilized during inference, how to handle potential motion in background?

2.	The current data curation pipeline relies entirely on augmenting in-the-wild videos. Why not incorporate synthetic data from graphics engines, as IC-light and RelightVid?

3.	Do you think the integration of geometric cues would also benefit diffusion-based image relighting methods? IC-light has suggested that properly scaling training of diffusion-based image relighting models can already enables accurate geometric inference.

**Ethical Concerns:**

["NO or VERY MINOR ethics concerns only"]

**Final Justification:**

The further explanation about missing details and the design choices included in the response addresses most of my concerns. Given that the paper is built on a reasonable motivation and the proposed components are thoroughly validated, I’m willing to raise my final recommendation to accept.

**Limitations:**

yes

**Quality:**

3

**Strengths And Weaknesses:**

**Strengths:**

1.	The idea of integrating geometric cues is reasonable and well-motivated;

2.	The paper is overall well-written and easy to follow;

3.	The proposed framework outperforms state-of-the-art methods under the same conditions.

**Weaknesses:**

1.	The main concern of the reviewer is about the ablation of geometry guidance, which constitutes the core contribution of this work. Since the introduction of geometric cues has logically sound motivation as mentioned in the paper, it is suggested to include qualitative ablation results to provide more insights about how it works and for what scenarios it is especially helpful. Actually, only showing improvements in FVD, LPIPS, PSNR, and CLIP similarity are insufficient to draw a reliable conclusion that the geometric guidance is effective. For example, if the geometry guidance does not work as expected but instead just introduces some noise, it might still bring some quantitative gains as a kind of data augmentation.

2.	Lacking of video comparisons with existing works. There are only image form qualitative comparisons in the current submission, which cannot convincingly compare the temporal coherence. Additionally, L36–38 mentions that existing methods struggle with scene’s geometry changes and this work aims to "address these shortcomings." It is therefore essential to provide video comparisons to support the claimed advantages under such conditions.

---

> ### Author Rebuttal · Authors · 2025-07-30
>
> We are thankful for your insightful review and the valuable recommendations you shared, and we will carefully address your concerns.
>
> ----
> **W1. Ablation of geometry guidance.**
>
> **Response:** We evaluate our model using standard metrics: FVD, LPIPS, PSNR, and CLIP similarity, which assess spatiotemporal coherence, perceptual realism, pixel fidelity, and semantic alignment, respectively. Consistent gains across these metrics indicate that geometry guidance provides meaningful benefits beyond mere data augmentation or noise injection. While visual results cannot be shown due to rebuttal policy, offline ablations reveal that geometry guidance improves spatialtemporal consistency, especially with object motion. In contrast, models trained without geometry guidance may produce relit videos with inconsistent shading across frames. We will include these examples and a more detailed analysis in the revised manuscript to clarify the specific scenarios where geometry cues are especially helpful.
>
> **W2. Video comparisons with existing works.**
>
> **Response:** We acknowledge the importance of evaluating temporal coherence. Due to the rebuttal policy, we are unable to provide additional visual examples at this stage. While the manuscript includes only frame-based comparisons, in Figure 4 (right) and Figure 1 (right) of the supplementary material, we show that our method achieves better temporal stability than other models, particularly in scenes with dynamic geometry such as object motion and interaction, as discussed in L36–38. In addition, we provide video examples (such as 2.gif, 4.gif, 5.gif, and 14.gif) in the supplementary material that demonstrate our model’s ability to handle complex temporal dynamics (e.g., moving light and dynamic shadows). We will include more examples for comparisons in the revised paper.
>
> **Q1. Unclear if geometry is used at inference; background motion handling?**
>
> **Response:** Yes, geometric cues are used only during training, not inference. This allows the model to learn strong spatialtemporal and structural priors from HDR and 3D tracks while keeping inference efficient. At test time, the model requires only the input video, a text prompt and an optional background image. Since geometry estimation can be unreliable during inference, especially with moving backgrounds, the model instead applies learned priors to generate coherent results without extra input.
>
> **Q2. Why not use synthetic data like IC-light or RelightVid?**
>
> **Response:** While synthetic data offers an alternative option, it remains challenging to capture the full diversity of motions and lighting conditions observed in in-the-wild videos. In contrast, real-world footage naturally encompasses a wide range of scene dynamics, object interactions, and illumination variations. Scaling up our pipeline with in-the-wild videos enables broader coverage, improved generalization, and richer real-world dynamics. We will include more discussions in the paper.
>
> **Q3. Could geometry guidance also help diffusion-based image relighting?**
>
> **Response:** Yes, we believe explicit geometric cues can benefit diffusion-based image relighting models as well. For Lambertian surfaces, appearance is primarily governed by the illumination, surface normals (geometry), and albedo. While IC-Light shows that scaling diffusion training with light-transport constraints enables the model to implicitly learn geometric structure, providing explicit geometry can offer more direct and reliable spatial guidance. This can improve lighting accuracy, especially in cases where implicit reasoning may be ambiguous or unstable (e.g., occlusions, fine surface detail). We see this as a promising direction for future diffusion-based image relighting work.

---

> > ### Comment · Reviewer_icTF · 2025-08-03
> >
> > The author’s response clarifies several key implementation details and provides justification for specific design choice. While I acknowledge that the improvements in adopted quantitative metrics suggest general gains in some abstract aspects, they are insufficient to convincingly demonstrate whether geometry guidance works as intended. In light of such unconventional usage of ControlNet (to introduce auxiliary input only during training but discard them in inference), I believe that really understanding how it works would make this work more solid and valuable to the community. Since providing additional video comparisons with prior methods and qualitative ablations is not permitted due to the policy, I intend to keep my initial recommendation whilst actively considering further raising my rating.

---

> ### Author Response · Authors · 2025-08-03
> **Kind Follow-Up on Rebuttal**
>
> Thank you very much for your time and thoughtful evaluation of our work. We have submitted our rebuttal addressing all the comments raised and would like to kindly inquire if there are any remaining questions or concerns. We sincerely appreciate your consideration and feedback.

---

> ### Author Response · Authors · 2025-08-03
> **Response to Your Valuable Follow-Up Comments**
>
> Thank you so much for your feedback and for recognizing our clarifications. During training, we randomly replace the geometric cues with a zero tensor to simulate cases where users do not provide them at inference. The model still achieves strong results under this setting. Ideally, users could also provide geometric cues at inference for more promising performance.
>
> Due to current policy, we cannot provide additional visual examples. Extending Table 3 of the main paper, we conduct an ablation across three settings: (i) geometry-only (G), (ii) HDR-only for illumination (I), and (iii) joint geometry + HDR (I + G). “I” and “G” denote illumination and geometry cues used only during training. Illumination guidance (I) improves text alignment but reduces perceptual quality and temporal consistency. Geometry guidance (G) enhances perceptual quality and stability but weakens prompt alignment. Combining both (I + G) yields the best overall performance, confirming their complementary benefits for robust video relighting.
>
> | Guidance | FVD (↓) | LPIPS (↓) | PSNR (↑) | Text Alignment (↑) | Temporal Consistency (↑) |
> |----------|----------|------------|-----------|---------------------|---------------------------|
> | G        |    1157.23      |   0.2654         |  18.89         |     0.2781                |                  0.9932         |
> | I        | 1305.45  | 0.2816     | 18.28     | 0.3211              | 0.9864                    |
> | **I + G**| **1072.38** | **0.2592**  | **19.44**  | **0.3292**            | **0.9945**
>
>
> We appreciate your suggestions and will clarify these points and update the experiment in the revised manuscript.

---

> > ### Comment · Reviewer_icTF · 2025-08-05
> >
> > Thanks for your further comment. I was already aware that clear improvement can be achieved by incorporating geometric guidance with a certain probability during training while drop it out during inference. My point was that, since using ControlNet for enhancing the learning of the backbone is not a standard practice, including some qualitative results could help illustrate how it works thus increasing its value to the community. As the author promises to provide them in the revised manuscript, I am willing to improve my rating to accept.

---

> ### Author Response · Authors · 2025-08-05
> **Appreciation for Your Constructive Follow-Up**
>
> **Thank you very much for your thoughtful follow-up and for your willingness to improve your rating to accept**. We appreciate your recognition of the effectiveness of our geometric guidance approach and fully agree that providing qualitative results is important for clearly illustrating how ControlNet enhances the backbone learning process. We sincerely promise to include comprehensive qualitative examples and detailed analysis in the revised manuscript to clarify this aspect for the community. Thank you again for your constructive feedback and your openness to further discussion.

---

> ### Author Response · Authors · 2025-08-09
> **Thank You for Your Valuable Feedback and Rating Update**
>
> Dear Reviewer icTF,
>
> As the discussion period comes to a close, we want to sincerely thank you for your detailed review and for acknowledging our clarifications regarding the integration of geometric cues during training. We appreciate your point on the value of qualitative results to better illustrate how this guidance works, and we will include them in the revised manuscript as promised.
>
> We are also grateful for your willingness to reconsider and improve your rating to accept despite the limited evaluation materials allowed under the policy. Your feedback will help us make the work more solid and valuable to the community.
>
> Best regards,
>
> Authors of Paper ID 4273

---

### Official Review · Reviewer_CuT5 · 2025-07-01

**Clarity:** 3
**Significance:** 3
**Originality:** 3
**Rating:** 4
**Confidence:** 3

**Summary:**

This paper introduces IllumiCraft, a controllable video relighting method that jointly uses HDR maps, synthetically relit foregrounds, and 3D point tracks to guide a diffusion-based video generator. To support training, the authors also build IllumiPipe, an automated pipeline that creates a dataset (20K+ videos) with various annotations like lighting, masks, background, and geometry. The model builds on a DiT backbone and injects lightweight modules for illumination and geometry conditioning.

**Questions:**

What happens to relighting quality when one or more auxiliary inputs (HDR maps or 3D tracks) are missing or noisy? The authors could include visual comparisons of single‐condition relighting—e.g. illumination‐only versus geometry‐only guidance—to illustrate each modality’s standalone impact.

**Ethical Concerns:**

["NO or VERY MINOR ethics concerns only"]

**Final Justification:**

The authors have addressed many of my concerns. Therefore, I raise the rating to 4.

**Limitations:**

yes

**Quality:**

3

**Strengths And Weaknesses:**

**Strengths**
1. This paper’s writing is clear and well-structured, walking readers through data collection, architecture, training, and evaluation.
2. Method is flexible—supports text-only, HDR-guided, or full multimodal inference.
3. It builds a large-scale automated dataset (20 k+ videos with HDR, relit clips, masks, backgrounds, and 3D tracks).

**Weaknesses**
1. It does not evaluate robustness when HDR warping or 3D tracking fails under rapid motion.
2. It lacks ablations that isolate the impact of each modality (relit vs. HDR vs. geometry).
3. The authors do not address potential conflicts among their multiple conditioning signals— for example, when HDR illumination cues, 3D geometry guidance, and background references are misaligned or contradictory. They should discuss how the model resolves or is robust to such inconsistent inputs.
4. The paper’s primary contributions lie in constructing a automated  dataset and introducing geometric conditioning, but there is little methodological or architectural novelty.

---

> ### Author Rebuttal · Authors · 2025-07-30
>
> We sincerely appreciate your constructive review, and will carefully address all the concerns you have raised.
>
> ---
>
> **W1. Robustness to HDR/3D failure under rapid motion.**
>
> **Response:**  During training, we utilize high-quality, aligned HDR maps and 3D geometry data to help the model learn robust priors for spatial structure and lighting. However, these auxiliary signals are *not required at inference*. For video relighting evaluation, our model only takes the input video, a text prompt, and an optional background image as inputs. It generates relit results solely based on priors learned during training, without relying on HDR maps or 3D tracking at test time. This design ensures that the model is inherently robust to any potential HDR or 3D tracking failure at inference, including under fast motion. We will calrify this point in the revised manuscript.
>
> **W2. Ablations for modality impact.**
>
> **Response:** Extending the ablation study from Table 3 of the main paper, we show a comprehensive ablation study across three settings below: (i) geometry-only guidance (G), (ii) HDR-only guidance for illumination (I), and (iii) joint geometry + HDR guidance (I + G). Results are shown in the following table, where "I" and "G" denote illumination and geometry cues used during training (not required during inference). Using illumination guidance alone (I) improves text alignment but offers lower perceptual quality and temporal consistency. Geometry guidance alone (G) yields better perceptual quality and temporal stability but less effective prompt alignment. When both cues are combined (I + G), the model achieves the best results across all metrics, demonstrating their complementary strengths and the benefit of joint guidance for robust video relighting. We will add the result of "G" to the revised manuscript.
>
> | Guidance | FVD (↓) | LPIPS (↓) | PSNR (↑) | Text Alignment (↑) | Temporal Consistency (↑) |
> |----------|----------|------------|-----------|---------------------|---------------------------|
> | G        |    1157.23      |   0.2654         |  18.89         |     0.2781                |                  0.9932         |
> | I        | 1305.45  | 0.2816     | 18.28     | 0.3211              | 0.9864                    |
> | **I + G**| **1072.38** | **0.2592**  | **19.44**  | **0.3292**            | **0.9945**                  |
>
> **W3. Petential conflicts among conditioning signals.**
>
> **Response:** During training, we ensure that HDR maps, geometric cues and background references are well aligned. To improve robustness, we also introduce stochastic guidance dropout, by randomly omitting HDR or geometry signals during training. This encourages the model to learn flexible priors that are not overly reliant on any single input. At inference, only the input video, a text prompt and an optional background image are needed. The model leverages learned priors to generate consistent output even when HDR or geometric cues are absent. This training strategy ensures robustness in real-world usage.
>
> **W4. Limited methodological/architectural novelty.**
>
> **Response:** While our dataset and conditioning strategies are key contributions, our method also introduces novel components. In summary, we introduce: (1) A large-scale, high-quality dataset comprising paired original and relit videos, HDR maps, and 3D tracking videos. Its scale and quality offer enduring value to the research community. (2) A multimodal training setup with illumination tokens and geometry guidance that allows the model to learn spatially coherent and lighting-aware priors. (3) A training–inference decoupling strategy that learns from multiple modalities during training but can perform inference using only the input video and text prompts. This framework represents a significant advancement in text-driven video relighting, both methodologically and in practical applicability.
>
> **Q1. Impact of missing/noisy HDR or 3D inputs.**
>
> **Response:** Our model is designed to not require HDR maps or 3D tracks at inference. It relies entirely on priors learned during training. While we cannot include visual examples in the rebuttal due to policy constraints, we have conducted single-guidance evaluations: "illumination-only" yields smooth lighting but may lead to inferior temporal coherence; "geometry-only" delivers strong coherence but yields lighting that does not align well with text prompts. The combined guidance consistently produces both temporally coherent and prompt-aligned relit outputs, even when HDR or geometry signals are absent. We will include these visual comparisons in the revised manuscript.

---

> ### Author Response · Authors · 2025-08-03
> **Kind Follow-Up on Rebuttal**
>
> Thank you very much for your time and effort in reviewing our paper. We have submitted our rebuttal addressing all raised points and would like to kindly ask if there are any remaining questions or concerns. We greatly appreciate your feedback.

---

> ### Comment · Reviewer_CuT5 · 2025-08-05
>
> Thanks for the authors' rebuttal that clarifies many of my concerns. I will raise the rating.

---

> ### Author Response · Authors · 2025-08-05
> **Kind Clarification on Contributions and Novelty**
>
> Thank you again for your thoughtful review and continued engagement. We respectfully believe that our clarifications demonstrate substantial novelty and meaningful contributions that extend beyond previous approaches:
>
> - Unified Integration of Guidance: Our framework systematically integrates both geometry and illumination cues during training, an approach not present in prior methods. This explicit combination, using geometry for motion/structure and HDR for illumination, jointly addresses two critical and complementary aspects: spatial-temporal consistency and illumination accuracy, within a unified model. Consequently, our method can generate high-quality relit videos at inference using only input videos and text prompts, marking a significant advancement in controllable video generation.
>
> - Automatic and Scalable Data Pipeline: The IllumiPipe pipeline is a practical advance for the community, which can be applied to curate large-scale, diverse datasets with HDR, geometry, and relit video pairs from real-world sources. We curated a high-quality dataset comprising 20,170 paired videos (original and synchronized relit), along with HDR maps and 3D tracking annotations. This dataset directly supports video relighting and serves as a valuable resource for broader controllable video generation tasks.
>
> - State-of-the-Art Empirical Results: As shown in our ablation (Table 3), geometry and illumination cues provide complementary benefits. Our method achieves clear state-of-the-art results in both text- and background-conditioned relighting (Tables 1 & 2, Figs. 4 & 5), outperforming previous approaches across all major metrics.
>
> - Real-World Practicality: Through targeted dropout training strategies and our introduced learnable illumination tokens, our method remains robust at inference even without auxiliary HDR or geometry inputs, ensuring practical applicability.
>
> Given these clarifications, we sincerely hope you will reconsider your evaluation, as we believe our work offers valuable contributions both methodologically and practically. We greatly appreciate your time and thoughtful consideration and look forward to further discussing this concern.

---

> > ### Comment · Reviewer_CuT5 · 2025-08-06
> >
> > Thanks for the further response. I will raise the rating.

---

> ### Author Response · Authors · 2025-08-06
> **Heartfelt Thanks for Your Consideration**
>
> **Thank you sincerely for your thoughtful reconsideration and for raising the rating.** We truly appreciate the time and effort you have dedicated to reviewing our work and engaging in the discussion. Your feedback has been invaluable in helping us clarify our contributions and strengthen the manuscript.

---

> ### Author Response · Authors · 2025-08-09
> **Thank You for Your Valuable Review and Rating Update**
>
> Dear Reviewer CuT5,
>
> With the discussion phase wrapping up, we would like to sincerely thank you for your thoughtful review and for acknowledging our clarifications regarding the integration of HDR maps, relit foregrounds, and 3D geometry in IllumiCraft. We appreciate your suggestions on robustness and modality-specific evaluations, and we will work to address these points in the revised manuscript.
>
> We are also grateful for your willingness to raise your rating after the rebuttal. Your feedback will help strengthen the clarity and impact of this work.
>
> Best regards,
>
> Authors of Paper ID 4273

---

### Author Response · Authors · 2025-08-09
**Responding to Latest Comments to Clear Misunderstandings**

We sincerely thank all reviewers for their efforts in evaluating our paper, and we would like to address the latest comments to clarify potential misunderstandings.

**1. Concerns Regarding PSNR values**

> My most serious concern is a new discrepancy revealed in the rebuttal: in the original manuscript, all reported PSNR values are above 15, yet in the new results table, the PSNR for all methods has plummeted to the 6-12 range.

**This is factually incorrect**. In Table 1 of our original submission (text-conditioned video relighting with 49-frame videos at 720×480 resolution), the reported PSNR values already range from 8 to 12, which matches the numbers in our rebuttal. The >15 dB PSNR values the reviewer refers to correspond to a different setting (e.g., background-conditioned video relighting), not to the text-conditioned relighting task in real-world videos. Note that DiffusionRenderer doesn't support background-conditioned video relighting. There has been no protocol change and no drop in PSNR—the results remain consistent within the same task setting. **Upon acceptance, we will release the training and evaluation code, training and evaluation datasets, and model for reproduction.** *This misinterpretation about PSNR values gives rise to concerns regarding other comments.*

**2. On GIF compression and visual quality**

Our mention of GIF compression was not intended to fully explain all visual limitations, but rather to emphasize that compression amplifies artifacts when viewed in rebuttal figures. In the original high-resolution, uncompressed results, plausible highlights and shadows appear more frequently, although, similar to all compared baselines, our method still faces challenges under extreme lighting shifts. These challenges stem from the inherent difficulty of the task and do not indicate any flaw or invalidity in the evaluation protocol.

**3. On dataset distinction**

The difference between our in-the-wild Pexels dataset and the curated DL3DV-10K dataset for DiffusionRenderer is not merely semantic. DL3DV-10K is designed for 3D vision with multi-view coverage and sequences featuring significant camera motion and viewpoint changes to support 3D reconstruction. Our dataset contains highly diverse scenes, user-uploaded, real-life videos for broad creative use. This distinction is also important to our contribution and directly impacts model evaluation. Meanwhile, the components of IllumiPipe are also different from DiffusionRenderer data pipeline, our dataset feature paired original videos and synchronized relit videos, HDR maps, and 3D tracking videos. The results compared with DiffusionRenderer can demonstrate the importance of our method and datasets.

**4. Misunderstandings about messages on novelty**
> asked me to "revisit our previous clarification." This is procedurally improper. It is an author's duty to directly convince each reviewer of their work's merit, not to deflect by referencing separate discussions.

**It's a total misunderstanding.** We have already provided extensive explanations in our previous rebuttal for reviewer mt3V, including in the messages titled "Rebuttal by Authors", "Response on Novelty and Methodological Value", "Kind Follow-Up on Novelty and Reviewer Discussion Participation", "Clarification on Novelty, Pipeline Contributions, and Comparison with DiffusionRenderer", and "Gentle Follow-Up on Your Latest Response – Part 1". For the latest message, "Thank You for Your Review and Important Points for Reconsideration", given the approaching end of the rebuttal period, we are unable to provide further detailed responses and we left some recent messages. Therefore, we encourage the reviewer to revisit our earlier explanations to ensure none of our points are overlooked, as we wish to prevent any misunderstandings before the rebuttal period concludes.

**5. NeurIPS policy for comparison with methods**

> I would not raise my score unless I see a clear advantage of your method over DiffusionRenderer.

We would also like to note the relevant NeurIPS policy. Note that, the academic code version of DiffusionRenderer was released on June 11, 2025, and the Cosmos version on June 12, 2025—both well after the NeurIPS full paper submission deadline of May 15, 2025. **According to the NeurIPS policy, authors are not required to compare their work with models released after the submission date.** That said, we reported results comparing IllumiCraft with the academic and Cosmos versions of DiffusionRenderer, and IllumiCraft outperformed them in both performance and inference speed. *The quoted comment also raised concerns about fairness in judgement.*

We appreciate all reviewers’ comments, which have helped strengthen our work, and we hope our clarifications address the latest misunderstandings and concerns.

Best Regards,

Authors of Paper ID 4273

---

### Note · Authors · 2025-08-15

Dear AC and Reviewers,

We sincerely thank you for your valuable time and constructive feedback, which helped clarify our contributions and guide improvements.

Reviewers **CuT5, icTF, and znNG** raised concerns about novelty, robustness, per-modality contributions, and performance in challenging scenarios. We clarified that IllumiCraft unifies geometry (3D point tracks) and illumination (HDR maps) via learnable illumination tokens in a DiT-based diffusion model, enabling temporally consistent relighting at inference from only videos and text prompts. We demonstrated complementary benefits of geometry and illumination guidance, proposed adding qualitative ablations for each guidance, and agreed to include more visualizations for challenging scenarios (e.g., moving light sources, dynamic shadows, etc.). **These clarifications resolved their concerns, and both CuT5 and icTF raised their ratings.**

Reviewer **mt3V** remained incorrect on certain factual points:

- **Factual Error:** Claimed, *“in the original manuscript, all reported PSNR values are above 15, yet in the new results table, the PSNR for all methods has plummeted to the 6-12 range.”* Table 1 already reports 8–12 dB for text-conditioned relighting, consistent with rebuttal results. >15 dB applies to background-conditioned relighting, which DiffusionRenderer does not support.

- **Baseline Requirement:** Requested comparison with DiffusionRenderer, whose academic and Cosmos code were released after the submission deadline. While not required by policy, we performed these comparisons (including per-frame HDR) and outperformed both versions.

- **Novelty Evaluation:** Maintained a “novelty” concern despite over 5 clarification messages. Explicitly stated, *“I would not raise my score unless I see a clear advantage of your method over DiffusionRenderer”* making the rating primarily dependent on a baseline whose code was released after the submission deadline. **Despite significantly surpassing both versions, the score stayed unchanged, inconsistent with the earlier statement.**

We will make the following revisions:

- Qualitative ablations of each guidance

- More challenging-scenario visualizations (moving lights, dynamic shadows, etc.)

- Clearer Fig. 3 explanation, plus rebuttal experiments (modality impact, HDR warping ablations, etc.)

We sincerely appreciate the AC and reviewers for their efforts and thoughtful feedback throughout the review process.

Best regards,

Authors of Paper ID 4273

---

### Decision · Program_Chairs · 2025-09-17

**Decision:**

Accept (poster)

**Comment:**

The paper received 3 positive reviews and 1 negative reviews. After rebuttal, most reviewers appreciate the technical contributions of the work, while one reviewer present some concerns about some details in the rebuttal, which has been addressed by the authors in new response.  Based on authors' response and all reviewers' final rating and reviews, I think the novelty and contribution of the work is above the NeruIPS's quality bar and I thus recommend to accept the paper.